# Measure Twice, Cut Once: A Semantic-Oriented Approach to Video Temporal Localization with Video LLMs

**Zongshang Pang**[1]    **Mayu Otani**[2]    **Yuta Nakashima**[1]
[1]The University of Osaka    [2]CyberAgent, Inc.
{pangzs,n-yuta}@im.sanken.osaka-u.ac.jp    otani_mayu@cyberagent.co.jp

## Abstract

Temporally localizing user-queried events through natural language is a crucial capability for video models. Recent methods predominantly adapt video LLMs to generate event boundary timestamps for temporal localization tasks, which struggle to leverage LLMs' pre-trained semantic understanding capabilities due to the uninformative nature of timestamp outputs. In this work, we explore a timestamp-free, semantic-oriented framework that fine-tunes video LLMs using two generative learning tasks and one discriminative learning task. We first introduce a structural token generation task that enables the video LLM to recognize the temporal structure of input videos based on the input query. Through this task, the video LLM generates a sequence of special tokens, called structural tokens, which partition the video into consecutive segments and categorize them as either target events or background transitions. To enhance precise recognition of event segments, we further propose a query-focused captioning task that enables the video LLM to extract fine-grained event semantics that can be effectively utilized by the structural tokens. Finally, we introduce a structural token grounding module driven by contrastive learning to associate each structural token with its corresponding video segment, achieving holistic temporal segmentation of the input video and readily yielding the target event segments for localization. Extensive experiments across diverse temporal localization tasks demonstrate that our proposed framework, MeCo, consistently outperforms methods relying on boundary timestamp generation, highlighting the potential of a semantic-driven approach for temporal localization with video LLMs [1].

## 1 Introduction

Localizing temporal events based on user interests is an essential capability for video recognition systems to handle practical video tasks such as moment retrieval (Lei et al., 2021), action localization (Chao et al., 2018; Cheng & Bertasius, 2022), video summarization (Song et al., 2015; Gygli et al., 2014), and dense video captioning (Krishna et al., 2017; Wang et al., 2021a; Yang et al., 2023). While such temporal localization tasks were traditionally handled by specialist models, recent research has begun leveraging video LLMs to unify them within a single framework by adapting LLMs for boundary timestamp generation (Ren et al., 2024; Liu et al., 2024c; Zeng et al., 2025).

Accurately localizing the boundary timestamps of a target event requires understanding the semantic content of the target event by both examining its relevance to the localization query (*e.g.*, an event description or action label) and differentiating it from adjacent events to identify event boundaries. However, current methods expect video LLMs to internally handle the semantic understanding and directly provide the final boundary timestamps. Consequently, a major focus of this line of research has been to develop various video LLM-friendly timestamp representations to boost performance. Nonetheless, direct timestamp generation may limit the potential of video LLMs, as they are backed by LLMs that are built to process semantic information (Brown et al., 2020) and have primarily been pre-trained on video captioning and question answering tasks that require mapping visual inputs to

---

[1]Code available at https://github.com/pangzss/MeCo.

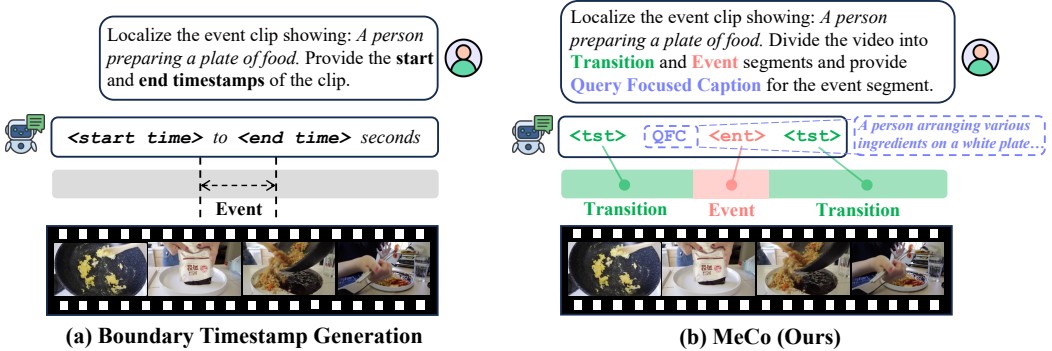

Figure 1: In contrast to previous boundary timestamp generation approaches (Ren et al., 2024; Guo et al., 2025a; Liu et al., 2024c; Huang et al., 2024; 2025; Guo et al., 2025b), MeCo leverages the semantic understanding capability of video LLMs to capture the video temporal structure and categorize video segments into transition and event structural tokens <tst> and <ent>. It can also generate query-focused captions (QFC) to scrutinize the detailed event semantics, which can facilitate more accurate localization of the event segment via the event token. The semantic-oriented pipeline completely frees video LLMs from timestamps generation.

outputs with concrete semantic meanings (Maaz et al., 2023; Lin et al., 2023a). Additionally, it has been shown that LLMs struggle with highly uninformative outputs in both language (Wei et al., 2022; Kojima et al., 2022) and multimodal scenarios (Lai et al., 2024; Liu et al., 2024c). In this work, we initiate the exploration of semantic-oriented strategies for adapting video LLMs to video temporal localization tasks. We adapt video LLMs' semantic understanding ability for such tasks through carefully designed supervised fine-tuning tasks involving both generative and discriminative learning objectives.

To start, we propose a *structural token generation* task to enable the video LLM to distinguish semantic differences between queried events and background transitions by capturing the overall temporal video structure. The output consists of consecutive segments in the video categorized as a sequence of *event tokens* <ent> and *transition tokens* <tst>, arranged in temporal order. To facilitate more precise categorization of event segments via event structural tokens, we propose a *query-focused captioning* task that enables the video LLM to scrutinize the details in each queried event before localization by generating detailed captions for it, akin to the role of Chain-of-Thoughts before LLMs' final answers (Wei et al., 2022; Kojima et al., 2022). These two tasks exploit the innate generative power of video LLMs to adapt their semantic understanding abilities for temporal localization tasks.

Building on the temporal categorization from structural tokens and the semantic refinement from query-focused captions, we propose a *structural token grounding* module based on contrastive learning (He et al., 2020; Radford et al., 2021; Wang et al., 2021b; Oquab et al., 2023; Pang et al., 2024) to map the rich semantics encoded in each structural token to the corresponding video segments. This enables holistic video segmentation, where queried events can be precisely localized through their corresponding structural tokens. The contrastive learning objective effectively taps into the hidden discriminative power of LLMs (BehnamGhader et al., 2024; Liu et al., 2024c) for temporal localization tasks.

The proposed framework, named MeCo, enables video LLMs to **Me**asure twice by prioritizing the semantic understanding of holistic video structure and fine-grained event content before **C**utting **o**nce to extract all queried event segments. This design fundamentally differs from timestamp generation-centric approaches, making MeCo a fully semantic-centric approach for video LLM-based temporal localization. An illustration of the three proposed objectives is shown in Fig. 1. Extensive experiments show that MeCo consistently outperforms timestamp-centric approaches, often by significant margins, across 9 tasks spanning grounding, dense video captioning, and complex reasoning domains (Liu et al., 2024c).

## 2    RELATED WORK

**Video Temporal Localization Tasks.** Video temporal localization tasks such as moment retrieval and highlight detection (Lei et al., 2021), extractive video summarization (Gygli et al., 2014; Song et al., 2015; Pang et al., 2023), and action localization (Chao et al., 2018; Cheng & Bertasius, 2022) require localizing salient event segments in response to a user query in the form of event boundary timestamps. Furthermore, tasks such as dense video captioning (Krishna et al., 2017; Yang et al., 2023) and grounded video question answering (Bärmann & Waibel, 2022; Di & Xie, 2024) involve generating captions and performing complex reasoning about these localized events. Traditionally, these tasks have been addressed by specialist models with task-specific designs and domain-specific training data. Although unified models for localization-only tasks have been proposed (Lin et al., 2023c; Yan et al., 2023; Liu et al., 2024b), they cannot handle generative tasks like captioning.

**Video LLMs.** Early efforts to enable LLMs to perform video-level tasks used LLMs as agents built on chain-of-thought reasoning and tool-use mechanisms (Zeng et al., 2022; Surís et al., 2023; Lin et al., 2023b). Advances in end-to-end multimodal pretraining (Radford et al., 2021; Li et al., 2023) and instruction tuning (Ouyang et al., 2022; Liu et al., 2024a) have led to the development of powerful video LLMs (Zhang et al., 2023; Song et al., 2024; Li et al., 2024; Wang et al., 2024b; Li et al., 2025; Yuan et al., 2025), which have have shown that these models excel at temporal reasoning over very long videos, benefiting from LLMs' long-context semantic retrieval abilities. However, while effective for general video understanding tasks such as captioning and question answering, they do not address tasks that require event temporal localization.

**Temporal Localization Video LLMs.** Recent developments in temporal localization video LLMs have enabled unified approaches for both localization and generation tasks. Models such as TimeChat (Ren et al., 2024) fine-tune pre-trained video LLMs to output numeric tokens that represent event boundary timestamps. Subsequent works (Huang et al., 2025; Qian et al., 2024; Guo et al., 2025a;b) augment the LLM's vocabulary with learnable timestamp tokens. For example, VTG-LLM represents timestamps with a set of learnable digit tokens and pads the timestamp token sequence to a fixed length (Guo et al., 2025a). Meanwhile, TRACE introduces specialized timestamp encoder and decoder (Guo et al., 2025b) and VideoChat-T proposes a teporal adaptive position encoding module (Zeng et al., 2025) . There are also works that directly interleave the textual timetsamps with frame tokens (Meinardus et al., 2024; Lu et al., 2024). Observing that LLMs struggle with numeric tokens and many newly introduced tokens, E.T. Chat (Liu et al., 2024c) instead fine-tunes LLMs on a boundary embedding matching task using a single boundary matching token. As a departure from such works relying on timestamp generation, we propose MeCo to explore how the innate semantic understanding capabilities of video LLMs can be leveraged to build a unified temporal localization framework.

## 3    METHOD

### 3.1    OVERVIEW

Video temporal localization involves understanding user-queried events and determining their temporal boundaries $\{(t_i^s, t_i^e)\}_{i=1}^M$ with $M \geq 1$. Depending on the query, the localized boundaries should sometimes be accompanied by event captions (Krishna et al., 2017) or answers to event-related questions (Bärmann & Waibel, 2022), which are treated as textual tokens $\{\boldsymbol{x}_n\}_{n=1}^N$, where $N$ is the total number of tokens. Recently, such temporal localization tasks have been unified within a single video LLM-based framework (Ren et al., 2024; Guo et al., 2025a).

Current video LLM-based methods focus on boundary timestamp generation, which fails to leverage video LLMs' core strength: their pre-trained semantic understanding capabilities. In contrast, we leverage video LLMs' semantic understanding and retrieval capabilities to partition the input video into segments, categorize them as target events or background transitions, and summarize their semantics in the hidden states of the proposed structural tokens. To ensure precise retrieval of event semantics, we augment the structural tokens with query-focused captions. Finally, the structural token grounding module maps the structural tokens to their corresponding video segments via the retrieved and summarized semantics in their hidden states. We now describe in detail how we equip the video LLM with these capabilities. An illustration of the proposed components is in Fig. 2.

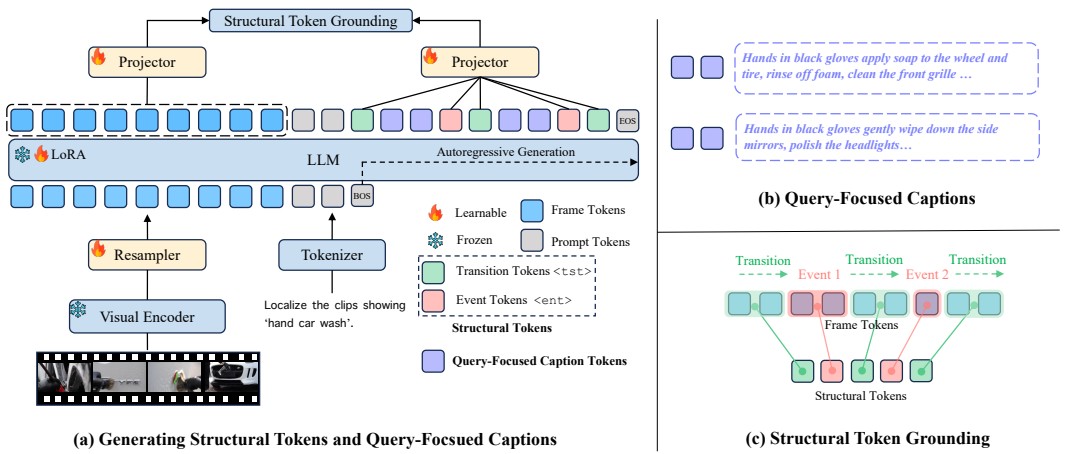

**(a) Generating Structural Tokens and Query-Focsued Captions**

**(b) Query-Focused Captions**

**(c) Structural Token Grounding**

Figure 2: Overview of the proposed MeCo framework. Given an input video and a localization-related user query, MeCo generates **structural tokens** (Sec. 3.2), including the event token `<ent>` and the transition token `<tst>`, enhanced by **query-focused captions** (Sec. 3.3) to encode more precise semantic information and can facilitate holistic temporal segmentation via **structural token grounding** (Sec. 3.4).

## 3.2 STRUCTURAL TOKEN GENERATION

Although video LLMs have demonstrated excellent temporal structure understanding for video question answering through causal reasoning and for video captioning by describing narrative flow (Song et al., 2024; Wang et al., 2024c; Zhang et al., 2024; Xue et al., 2024), this capability has not yet been explicitly leveraged for temporal localization. To fill this gap, we propose a novel structural token generation task for training the video LLM to materialize its temporal structure understanding into a temporally ordered sequence of video segments to facilitate temporal localization.

Given a $T$-frame video as input[2], a visual encoder (Fang et al., 2023b) and a resampler (Liu et al., 2024c) from the video LLM extract from the video a set of frame feature maps $\{\boldsymbol{F}_t\}_{t=1}^T$, where $\boldsymbol{F}_t \in \mathbb{R}^{P \times C}$ has $P$ token embedding vectors, each of dimension $C$. An LLM decoder then takes $\{\boldsymbol{F}_t\}_{t=1}^T$ and the tokenized user query $\{\boldsymbol{q}_l\}_{l=1}^L$ as inputs. The structural token generation task requires the video LLM to distinguish between event segments and background transition segments in the input video based on the user query. The segments are then represented in the LLM's autoregressively generated output as either *event tokens* `<ent>` or *transition tokens* `<tst>`, collectively called *structural tokens*, which are newly added to the LLM vocabulary before fine-tuning.

The preparation of supervised fine-tuning data for the structural token generation task builds on ground-truth event boundary timestamps from general temporal localization data. Given a $T$-frame video with a set of $M$ ground-truth event boundary timestamps $\{(t_i^s, t_i^e)\}_{i=1}^M$, we

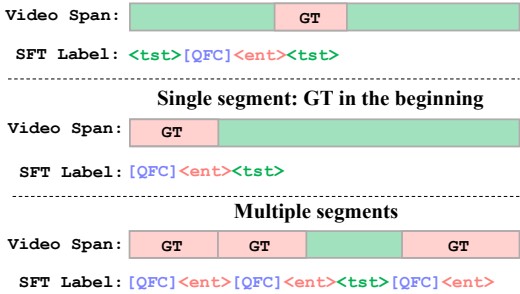

Figure 3: Examples of supervised fine-tuning (SFT) labels created from temporal localization data, where GT stands for Ground Truth windows and QFC for Query-Focused Caption (Sec. 3.3). For illustration clarify, we only show non-overlapping events for the multi-segment scenario, as overlapping events can be represented in a way similar to consecutive events, *e.g.*, `[QFC]<ent>[QFC]<ent>`.

augment them with neighboring transition segments to form an augmented set of segments $\{(t_i^s, t_i^e)\}_{i=1}^{M+K}$, where $K$ is the total number of transition segments. Let $\mathbb{I}_{\text{ent}}$ be the set of indices of

---

[2]By default, we sample frames from videos at 1 fps unless specified otherwise.

the queried event segments; the sequence of structural tokens can be defined as $\{\text{ST}(i)\}_{i=1}^{M+K}$ with

$$\text{ST}(i) = \begin{cases} \texttt{<ent>} & \text{if } i \in \mathbb{I}_{\text{ent}}, \\ \texttt{<tst>} & \text{otherwise}. \end{cases} \tag{1}$$

Importantly, event segments may appear at video boundaries or occur consecutively without intervening transitions. These cases provide crucial learning signals: the `<tst>` token does not always trivially precede and follow the `<ent>` token, and should only appear based on the actual presence of transition segments in the video. Several illustrative examples are provided in Fig. 3. Overall, structural tokens transform the video's temporal flow into a sequence of events and transitions. Through auto-regressive generation, each structural token must attend to its corresponding segment, thereby summarizing the segment's semantic information in its hidden state. This creates the foundation for grounding structural tokens to their corresponding video segments for localization purposes.

### 3.3 QUERY-FOCUSED CAPTIONING

Just as humans rewatch clips to identify specific details of interest, we hypothesize that the summarized segment semantics encoded in structural tokens can be refined by having the LLM examine each event segment more closely. To this end, we introduce a query-focused captioning task that requires the model to generate detailed captions for queried segments. By generating these captions immediately before each corresponding event token `<ent>`, we provide rich semantic information that the token can attend to via causal attention. This process mirrors chain-of-thought reasoning (Wei et al., 2022), where intermediate reasoning steps enhance the quality of final outputs, but resorts to more obtainable captions.

As shown in Fig. 2 and Fig. 3, the overall tokens that the LLM needs to generate now become an interleaved sequence of structural tokens and query-focused caption tokens $\boldsymbol{X} = \{\text{CAP}(i), \text{ST}(i)\}_{i=1}^{M+K}$ with

$$\text{CAP}(i) = \begin{cases} [\texttt{QFC}]_i & \text{if } i \in \mathbb{I}_{\text{ent}}, \\ \varnothing & \text{otherwise}, \end{cases} \tag{2}$$

where $[\texttt{QFC}]_i$ encloses all query-focused caption tokens for the $i$-th event segment, and $\varnothing$ indicates no such token is placed (we omit the end-of-sequence token for notational clarity). It is worth noting that textual response tokens for tasks involving captioning and question answering are treated as part of the query-focused caption tokens to unify the output format across all temporal localization tasks. The training objectives of structural token generation and query-focused captioning can be unified as a single language modeling loss:

$$\mathcal{L}_{\text{LM}} = -\frac{1}{N} \sum_{n=1}^{N} \log p(\boldsymbol{X}_n | \{\boldsymbol{F}_t\}_{t=1}^{T}, \{\boldsymbol{q}_l\}_{l=1}^{L}, \boldsymbol{X}_{<n}), \tag{3}$$

where $\boldsymbol{X}_n$ is the $n$-th token in $\boldsymbol{X}$, $\boldsymbol{X}_{<n} = \{\boldsymbol{X}_{n'}\}_{n'=1}^{n-1}$, and $N$ is the total number of tokens in $\boldsymbol{X}$.

As query-focused captioning is a novel task with no existing dataset, we leverage the ground-truth event timestamps in E.T.Instruct (Liu et al., 2024c) to extract event clips, which are then sent to a video captioning model to generate detailed clip captions. Further details regarding the generation pipeline are presented in the Appendix C.

### 3.4 STRUCTURAL TOKEN GROUNDING

To ground the generated structural tokens to their corresponding video segments, we leverage LLM hidden states to maximize the log-likelihood of the structural tokens with respect to their corresponding segment frames. Formally, given the projected LLM hidden states $\{\boldsymbol{H}_t\}_{t=1}^{T}$ and $\{\mathbf{s}_i\}_{i=1}^{M+K}$ (from two learnable MLP projectors following Liu et al. (2024c)) of the video frames and the structural tokens, respectively, where $\boldsymbol{H}_t \in \mathbb{R}^{P \times C}$ and $\mathbf{s}_i \in \mathbb{R}^{C}$, the structural token grounding loss can be formulated as:

$$\mathcal{L}_{\text{ST}} = -\frac{1}{M+K} \sum_{i=1}^{M+K} \sum_{t=t_i^s}^{t_i^e} \frac{\log p(\boldsymbol{h}_t | \mathbf{s}_i)}{t_i^e - t_i^s}, \tag{4}$$

where $\boldsymbol{h}_t \in \mathbb{R}^C$ is spatially average-pooled from $\boldsymbol{H}_t$, $\tau$ is a learnable temperature parameter (Radford et al., 2021), and both $\boldsymbol{h}_t$ and $\mathbf{s}_i$ are normalized to the unit sphere following (Radford et al., 2021; He et al., 2020). The conditional probability $p(\boldsymbol{h}_t|\mathbf{s}_i)$ of frame $t$ given structural token $i$ is computed as:

$$p(\boldsymbol{h}_t|\mathbf{s}_i) = \frac{\exp(\mathbf{s}_i \cdot \boldsymbol{h}_t / \tau)}{\sum_{t'=1}^{T} \exp(\mathbf{s}_i \cdot \boldsymbol{h}_{t'} / \tau)}, \tag{5}$$

which essentially makes Eq. (4) a contrastive learning objective (He et al., 2020; Radford et al., 2021) that pulls together structural tokens and their corresponding segment frames. We also attempted the symmetric version of Eq. (4) by including $p(\mathbf{s}_i|\boldsymbol{h}_t)$, similar to Radford et al. (2021), but observed performance drops, for which we provide analysis in Sec. 4.4, and thus we choose Eq. (4) as the final loss function.

### 3.5 TRAINING AND INFERENCE

The overall training objective of the video LLM is a combination of the language modeling loss and the structural token grounding loss:

$$\mathcal{L} = \mathcal{L}_{\text{LM}} + \mathcal{L}_{\text{ST}}, \tag{6}$$

where $\mathcal{L}_{\text{LM}}$ and $\mathcal{L}_{\text{ST}}$ are defined in Eq. (3) and Eq. (4), respectively. We use the E.T. Instruct dataset (Liu et al., 2024c) containing 164K samples augmented with query-focused captions and updated instructions as the supervised fine-tuning dataset. More details regarding the dataset and instructions are provided in Appendix C and Appendix G.

During inference, the video LLM recognizes the video's temporal structure to determine the presence of event and transition segments, and then auto-regressively generates corresponding structural tokens (`<ent>` and `<tst>`). Before generating each `<ent>` token, the model first produces a query-focused caption for that event segment. Upon completion of structural token generation indicated by the end-of-sentence token `<EOS>`, we compute $p(\boldsymbol{h}_t|\mathbf{s}_i)$ as in Eq. (5) for all frames with respect to each structural token. We then obtain holistic temporal segmentation by assigning each frame to the structural token that yields the highest conditional probability. This directly yields the queried event segments via the event tokens `<ent>`. The pseudocode for MeCo inference is provided in Appendix E.

## 4 EXPERIMENTS

### 4.1 BENCHMARKS

We evaluate MeCo's zero-shot temporal localization performance on three benchmarks: E.T. Bench (Liu et al., 2024c), Charades-STA (Gao et al., 2017), and QVHighlights (Lei et al., 2021).

**E.T. Bench** is a comprehensive benchmark composed of curated data from various public benchmarks for event-level and time-sensitive video tasks. We utilize the grounding, dense captioning, and complex temporal reasoning domains in E.T. Bench for evaluating our method. The grounding domain, with F1 score as the evaluation metric, includes five tasks: Temporal Video Grounding (`TVG`) (Lei et al., 2021; Gao et al., 2017), Episodic Memory (`EPM`) (Grauman et al., 2022), Temporal Action Localization (`TAL`) (Patraucean et al., 2023; Gorban et al., 2015; Jiang et al., 2014), Extractive Video Summarization (`EVS`) (Song et al., 2015; Gygli et al., 2014), and Video Highlight Detection (`VHD`) (Song et al., 2015; Gygli et al., 2014). The dense captioning domain, with F1 score and sentence similarity as the evaluation metrics, consists of two tasks: Dense Video Captioning (`DVC`) (Zala et al., 2023; Zhou et al., 2018) and Step Localization and Captioning (`SLC`) (Zhukov et al., 2019; Afouras et al., 2023). The complex temporal reasoning domain, with recall as the evaluation metric, includes two tasks: Temporal Event Matching (`TEM`) (Patraucean et al., 2023; Lei et al., 2021) and Grounded Video Question Answering (`GVQ`) (Bärmann & Waibel, 2022).

**Charades-STA and QVHighlights** are widely adopted benchmarks for temporal grounding and video highlight detection. The `TVG` task of E.T. Bench contains samples from both these two benchmarks, but does not evaluate the highlight detection performance for QVHighlights. We thus report additional results directly on the original benchmarks to facilitate comparisons with existing methods. For Charades-STA (Gao et al., 2017), we use recall at temporal Intersection over Union thresh-

Table 1: Zero-shot performance comparisons on E.T. Bench. "SFT. Data" refers to the supervised fine-tuning data used in the temporal localization tuning stage, which may include both localization-specific and other video data. Grayed out metrics are not zero-shot results as the model accessed the training data of the corresponding evaluation data in E.T. Bench. **Bold** and *italic* only are used for comparisons in the self-implemented E.T.Instruct fine-tuning-based setting.

| Model | SFT Data | Epochs | LoRA | Grounding | | | | | Dense Captioning | | | | Complex | |
|---|---|---|---|---|---|---|---|---|---|---|---|---|---|---|
| | | | | $TVG_{F1}$ | $EPM_{F1}$ | $TAL_{F1}$ | $EVS_{F1}$ | $VHD_{F1}$ | $DVC_{F1}$ | $DVC_{Sim}$ | $SLC_{F1}$ | $SLC_{Sim}$ | $TEM_{Rec}$ | $GVQ_{Rec}$ |
| *Video LLMs prompted with timestamps by Liu et al. (2024c)* | | | | | | | | | | | | | | |
| Video-ChatGPT (7B) (Maaz et al., 2023) | | | | 7.0 | 1.3 | 15.1 | 8.4 | 28.8 | 8.8 | 11.3 | 5.7 | 10.2 | 15.9 | 0.0 |
| PLLaVA (7B) (Xu et al., 2024) | - | - | - | 6.9 | 1.1 | 5.7 | 0.3 | 28.9 | 13.3 | 10.6 | 9.7 | 11.8 | 4.1 | 1.2 |
| Video-LLaVA (7B) (Lin et al., 2023a) | | | | 7.0 | 1.9 | 15.0 | 0.3 | 28.9 | 28.0 | 15.0 | 0.9 | 8.3 | 7.5 | 0.1 |
| Video-LLaMA-2 (7B) (Zhang et al., 2023) | | | | 0.1 | 0.0 | 0.0 | 0.0 | 1.5 | 0.6 | 14.5 | 0.0 | 15.2 | 0.0 | 0.1 |
| *Temporal Localization video LLMs trained with their respective settings.* | | | | | | | | | | | | | | |
| VTimeLLM (7B) (Huang et al., 2024) | 142K | 2 | ✓ | 7.6 | 1.9 | 18.2 | 15.9 | 28.9 | 12.4 | 13.1 | 8.7 | 6.4 | 6.8 | 1.9 |
| VTG-LLM (7B) (Guo et al., 2025a) | 217K | 10 | ✓ | 15.9 | 3.7 | 14.4 | 26.8 | 48.2 | 40.2 | 18.6 | 20.8 | 14.4 | 8.9 | 1.4 |
| TimeChat (7B) (Ren et al., 2024) | 198K | 3 | ✓ | 26.2 | 3.9 | 10.1 | 29.1 | 40.5 | 16.6 | 12.5 | 5.6 | 9.2 | 18.0 | 1.5 |
| LITA (13B) (Huang et al., 2025) | 500K | 1 | × | 22.2 | 4.6 | 18.0 | 29.7 | 23.9 | 39.7 | 17.2 | 21.0 | 12.2 | 16.0 | 2.2 |
| TRACE (7B) (Guo et al., 2025a) | 900K | 2 | × | 44.3 | 11.1 | 19.1 | 27.4 | 65.9 | 46.4 | 21.7 | 28.3 | 18.1 | 19.2 | 0.0 |
| Dispider (7b) (Qian et al., 2025) | 639K | 1 | × | 43.6 | 17.2 | 29.9 | - | 51.5 | 31.6 | 17.8 | 14.1 | 11.7 | - | - |
| E.T.Chat (3.8B) (Liu et al., 2024c) | 164K | 1 | ✓ | 38.6 | 10.2 | 30.8 | 25.4 | 62.5 | 38.4 | 19.7 | 24.4 | 14.6 | 16.5 | 3.7 |
| *Fine-tuned on E.T.Instruct by Yuan et al. (2025)* | | | | | | | | | | | | | | |
| Tarsier (7B) (Wang et al., 2024a) | | | | 39.6 | 9.0 | 25.0 | 25.4 | 47.6 | 42.8 | 19.1 | 23.7 | 15.2 | - | - |
| Tarsier2 (7B) (Yuan et al., 2025) | - | - | - | 38.4 | 11.0 | 31.8 | 19.4 | 66.8 | 46.5 | 28.8 | 24.6 | 16.4 | - | - |
| QWen2-VL (7B) (Wang et al., 2024b) | | | | 39.7 | 7.0 | 26.9 | 17.1 | 66.9 | 44.3 | 25.3 | 25.7 | 15.6 | - | - |
| *Initialized from the last pre-trained checkpoint without seeing localization data and fine-tuned on E.T.Instruct. Self-implemented.* | | | | | | | | | | | | | | |
| VTG-LLM (7B) (Guo et al., 2025a) | | | | 20.0 | 1.5 | 17.6 | 19.5 | 41.6 | 39.9 | 19.6 | 19.6 | 13.6 | 18.2 | 0.3 |
| TimeChat (7B) (Ren et al., 2024) | | | | 24.3 | 2.3 | 17.7 | 23.6 | 43.0 | 39.4 | 16.5 | 18.0 | 11.8 | *19.1* | 0.8 |
| TRACE (7B) (Guo et al., 2025a) | | | | 18.5 | 2.2 | 22.3 | 26.7 | 38.2 | 39.0 | 17.5 | 23.4 | 13.7 | 12.5 | 1.4 |
| E.T.Chat (3.8B) (Liu et al., 2024c) | 164K | 1 | ✓ | 38.6 | 10.2 | 30.8 | 25.4 | 62.5 | 38.4 | 19.7 | 24.4 | 14.6 | 16.5 | 3.7 |
| **MeCo** (ETChat 3.8B) | | | | 59.1 | 11.2 | 32.6 | 33.2 | **66.9** | **43.4** | 20.3 | 27.3 | 15.7 | **23.6** | 9.6 |
| **MeCo** (ETChat 7B) | | | | **62.5** | 15.4 | **35.1** | 35.1 | 66.3 | **43.4** | 20.7 | **30.1** | **16.5** | 19.1 | 9.9 |
| **MeCo** (QWen2VL 7B) | | | | 59.0 | **17.5** | 34.2 | **35.5** | **67.9** | 41.5 | **22.8** | 28.1 | **16.5** | 15.4 | **15.1** |

olds of 0.3 (R@$1_{0.3}$), 0.5 (R@$1_{0.5}$) and 0.7 (R@$1_{0.7}$). For QVHighlights, we use mean Average Precision (mAP) and HIT@1 for highlight detection.

## 4.2 IMPLEMENTATION DETAILS

We develop MeCo using the E.T.Chat (Liu et al., 2024c) QWen2VL-7B (Wang et al., 2024b). For ETChat, we use its original Phi-3-Mini-3.8B (Abdin et al., 2024) version and implement a 7B version based on QWen2-7B (Yao et al., 2024) by following ETChat's three-stage training protocol, where the final stage involves temporal localization fine-tuning with LoRA (Hu et al., 2021) for one epoch. For QWen2VL-7B, we directly fine-tune the pre-trained checkpoint plus the newly initialized projectors from ETChat with the temporal localization data. More implementation details are provided in Appendix B.

## 4.3 MAIN RESULTS

**E.T. Bench: Comprehensive comparisons.** As shown in Tab. 1, although previous temporal localization video LLMs demonstrate promising zero-shot results compared to general video LLMs, they underperform on most tasks compared to MeCo. Specifically, MeCo (3.8B) achieves substantial gains across all domains. Notably, many models use larger base LLMs and train for considerably more steps. The results demonstrate that MeCo leverages video LLMs' semantic understanding more effectively for temporal localization than boundary-centric methods. Furthermore, when MeCo uses a more powerful base LLM (7B), its performance consistently improves on most tasks, reinforcing its scalability and effectiveness.

**E.T. Bench: Comparisons with E.T.Instruct fine-tuning.** When fine-tuned on E.T.Instruct, VTG-LLM (Guo et al., 2025a) and TimeChat (Ren et al., 2024) retain performance levels comparable to their original settings. TRACE (Guo et al., 2025b) experiences the greatest performance drop, likely because its specialized timestamp encoder/decoder and newly introduced timestamp tokens require extensive tuning for effective adaptation to the LLM. In contrast, MeCo emphasizes leveraging the inherent semantic understanding of video LLMs, making it more amenable to efficient fine-tuning.

**Temporal Grounding.** As shown in Tab. 2, MeCo achieves consistently better zero-shot performance on Charades-STA compared to either previous methods' official checkpoints or E.T.Instruct-fine-tuned checkpoints. After fine-tuning on the training set of Charades-STA, MeCo achieves significantly better results and retain the best performance for R@$1_{0.3}$ and R@$1_{0.5}$. However, MeCo prioritizes capturing general semantic differences between query-relevant and background frames

Table 2: Zero-shot and dataset-wise fine-tuning performance on Charades-STA and QVHighlights.

| Model | Charades-STA | | | QVHighlights | |
|---|---|---|---|---|---|
| | $R@1_{0.3}$ | $R@1_{0.5}$ | $R@1_{0.7}$ | mAP | HIT@1 |
| *Zero-shot performance (official checkpoints).* | | | | | |
| VTimeLLM (7B) (Huang et al., 2024) | 51.0 | 27.5 | 11.4 | - | - |
| VTimeLLM (13B) (Huang et al., 2024) | 55.3 | 34.3 | 14.7 | - | - |
| Momentor (7B) (Qian et al., 2024) | 42.6 | 26.6 | 11.6 | 7.6 | - |
| HawkEye (7B) (Wang et al., 2024d) | 50.6 | 31.4 | 4.5 | - | - |
| TimeChat (7B) (Ren et al., 2024) | - | 32.2 | 13.4 | 14.5 | 23.9 |
| VTG-LLM (7B) (Guo et al., 2025a) | - | 33.8 | 15.7 | 16.5 | 33.5 |
| TRACE (7B) (Guo et al., 2025b) | - | 40.3 | 19.4 | 26.8 | 42.7 |
| E.T.Chat (3.8B) (Liu et al., 2024c) | 64.4 | 43.2 | 19.4 | 23.2 | 58.9 |
| Seq2Time (7B) (Deng et al., 2025) | - | 31.2 | 13.7 | - | - |
| NumPro-FT (7B) (Wu et al., 2025) | 63.8 | 42.0 | 20.6 | 25.0 | 37.2 |
| VideoChat-T (7B) (Zeng et al., 2025) | 69.9 | 48.7 | 24.0 | 26.5 | 54.1 |
| *Zero-shot performance (Fine-tuned on E.T.Instruct).* | | | | | |
| TimeChat (7B) (Ren et al., 2024) | 43.4 | 24.9 | 9.2 | 16.4 | 30.6 |
| VTGLLM (7B) (Guo et al., 2025a) | 24.8 | 9.8 | 3.5 | 15.9 | 27.1 |
| TRACE (7B) (Guo et al., 2025b) | 39.4 | 23.7 | 11.5 | 16.4 | 30.6 |
| E.T.Chat (3.8B) (Liu et al., 2024c) | 64.4 | 43.2 | 19.4 | 23.2 | 58.9 |
| **MeCo** (ETChat 3.8B) | 66.7 | 44.4 | 17.5 | 39.2 | 61.8 |
| **MeCo** (ETChat 7B) | 69.6 | 46.4 | 19.1 | **39.5** | **64.3** |
| **MeCo** (QWen2VL 7B) | **71.1** | **50.1** | **23.3** | 37.2 | 57.9 |
| *Dataset-wise fine-tuning performance.* | | | | | |
| M-DETR (Lei et al., 2021) | 65.8 | 52.1 | 30.6 | 35.7 | 55.6 |
| UMT (Liu et al., 2022) | - | - | - | 39.9 | 64.2 |
| QD-DETR (Moon et al., 2023b) | - | 57.3 | 32.6 | 39.1 | 63.0 |
| CG-DETR (Moon et al., 2023a) | 70.4 | 58.4 | 36.3 | 40.8 | 66.7 |
| UniVTG (Lin et al., 2023c) | 72.6 | 60.2 | 38.6 | 38.8 | 61.8 |
| HawkEye (7B) (Wang et al., 2024d) | 72.5 | 58.3 | 28.8 | - | - |
| TimeChat (7B) (Ren et al., 2024) | - | 46.7 | 23.7 | 21.7 | 37.9 |
| VTG-LLM (7B) (Guo et al., 2025a) | - | 57.2 | 33.4 | - | - |
| TRACE (7B) (Guo et al., 2025b) | - | 61.7 | 41.4 | - | - |
| VideoChat-T (7B) (Zeng et al., 2025) | 79.4 | 67.1 | **43.0** | 27.0 | 55.3 |
| **MeCo** (ETChat 3.8B) | 75.3 | 61.6 | 38.5 | 44.6 | 71.8 |
| **MeCo** (ETChat 7B) | 77.2 | 63.9 | 40.1 | 44.7 | 74.3 |
| **MeCo** (QWen2VL 7B) | **82.3** | **68.5** | 41.6 | **45.3** | **75.1** |

rather than modeling dataset-specific phase-in and phase-out boundary patterns, it may achieve less impressive gains in terms of $R@1_{0.7}$.

**Highlight detection.** As the continuous semantic similarities derived from Eq. (5) can directly be utlized as highlight scores, MeCo achieves much higher performance in mAP and HIT@1 for highlight detection than previous methods, most of which generate numeric tokens to approximate highlight scores which struggle to capture the underlying semantic information. Impressively, fine-tuning with the training set of QVHighlights significantly improves MeCo's performance, which even prominently surpasses that of the specialist models. Importantly, MeCo is the only method that achieves a decent balance between temporal grounding and highlight detection.

## 4.4 ABLATION STUDY

In this section, all experiments are conducted with MeCo (3.8B) unless otherwise specified, and all metrics are reported as the average across all tasks within the corresponding domain in E.T. Bench.

**Semantic-based methods excel; video LLMs amplify.** In Tab. 3, we compare contrastive vision language models with temporal localization video LLMs. For each contrastive model, we compute the cosine similarities between the localization query feature and the frame features (sampled at 1 fps). We then apply a threshold to these similarity scores and merge contiguous points above the threshold as localized segments. Contrastive models built on semantic similarities perform impressively on grounding tasks without additional training. This provides solid proof for the strength of

Table 3: Comparisons between contrastive vision–language models and video LLMs.

| Model | $TVG_{F1}$ | $EPM_{F1}$ | $TAL_{F1}$ | $EVS_{F1}$ | $VHD_{F1}$ |
|---|---|---|---|---|---|
| *Contrastive Vision and Language Models* | | | | | |
| CLIP-L-14-224 (Radford et al., 2021) | 35.1 | 10.0 | 19.9 | 30.2 | 62.2 |
| EVA-G-14-224 (Fang et al., 2023a) | 39.7 | 12.7 | 21.7 | 31.4 | 61.8 |
| SIGLIP-L-16-384 (Zhai et al., 2023) | 42.5 | 14.1 | 22.5 | 29.8 | 63.4 |
| *Temporal Localization video LLMs* | | | | | |
| Previous best | 44.3 | 11.1 | 30.8 | 29.7 | 65.9 |
| MeCo (7B) | 62.5 | 15.4 | 35.1 | 35.1 | 66.3 |

Table 4: Comparisons with boundary-centric methods using the same base video LLMs and frame sampling strategies.

| Model | #Frames | $F1_{gnd}$ | $F1_{cap}$ | $Sim_{cap}$ | $Rec_{com}$ |
|---|---|---|---|---|---|
| *VideoLLaMa (7B)* | | | | | |
| + TimeChat (Ren et al., 2024) | | 22.2 | 28.7 | 14.1 | 9.9 |
| + VTGLLM (Guo et al., 2025a) | 96 | 20.0 | 29.7 | 16.6 | 9.3 |
| + MeCo | | **34.3** | **30.2** | **17.0** | **15.7** |
| *E.T.Chat Stage-2 (3B)* | | | | | |
| + E.T.Chat (Liu et al., 2024c) | | 33.5 | 29.1 | 16.3 | 8.6 |
| + MeCo | 1 fps | **40.6** | **35.4** | **18.0** | **16.6** |

Table 5: The necessity of `<tst>` token and query-focused captioning (QFC).

| Method | $F1_{gnd}$ | $F1_{cap}$ | $Sim_{cap}$ | $Rec_{com}$ |
|---|---|---|---|---|
| `<ent>` | 26.7 | 15.0 | 14.2 | 9.4 |
| `<ent>` + `<tst>` | 38.1 | 33.8 | **20.5** | 14.5 |
| `<ent>` + QFC | 40.4 | 32.0 | 19.9 | 14.9 |
| `<ent>` + Query Copying | 26.6 | 15.2 | 14.3 | 9.5 |
| `<ent>` + `<tst>` + QFC | **40.6** | **35.4** | 20.3 | **16.6** |

Table 6: The effect of using different variants of the structural token grounding loss $\mathcal{L}_{ST}$.

| $\mathcal{L}_{ST}$ Variant | $F1_{gnd}$ | $F1_{cap}$ | $Sim_{cap}$ | $Rec_{com}$ |
|---|---|---|---|---|
| $\mathcal{L}_{ST}(p(\boldsymbol{h}_t\|\mathbf{s}_i))$ | **40.6** | **35.4** | **20.3** | **16.6** |
| $\mathcal{L}_{ST}(p(\boldsymbol{h}_t\|\mathbf{s}_i)) + \mathcal{L}_{ST}(p(\mathbf{s}_i\|\boldsymbol{h}_t))$ | 39.9 | 33.4 | 15.9 | 15.2 |
| $\mathcal{L}_{ST}(p(\boldsymbol{h}_i^{seg}\|\mathbf{s}_i))$ | 23.2 | 23.5 | 12.9 | 13.4 |
| $\mathcal{L}_{ST}(p(\boldsymbol{h}_i^{seg}\|\mathbf{s}_i)) + \mathcal{L}_{ST}(p(\mathbf{s}_i\|\boldsymbol{h}_i^{seg}))$ | 24.3 | 25.9 | 13.6 | 13.4 |

Table 7: Investigation on the compatibility of QFC and different boundary-centric localization strategies, where Positional Embedding is from Ren et al. (2024), Interleaving from Meinardus et al. (2024), and Boundary Matching from Liu et al. (2024c).

| Loc. Strategy | $F1_{gnd}$ | $F1_{cap}$ | $Sim_{cap}$ | $Rec_{com}$ |
|---|---|---|---|---|
| Positional Embedding | 22.8 | **16.2** | **14.0** | **8.7** |
| + QFC | 14.8 | 15.1 | 11.8 | 8.0 |
| Interleaving | **27.1** | **31.8** | **15.9** | **12.4** |
| + QFC | 23.2 | 20 | 14.6 | 6.1 |
| Boundary Matching | **33.5** | **29.1** | 16.3 | **8.6** |
| + QFC | 30.5 | 26.9 | **18.9** | 7.9 |
| **Structural Tokens** | 38.1 | 33.8 | **20.5** | 14.5 |
| + QFC | **40.6** | **35.4** | 20.3 | **16.6** |

semantic-based approaches in temporal localization. By harnessing video LLMs' powerful semantic understanding capabilities, MeCo further amplifies this strength.

**Replacing boundary-centric methods with MeCo yields consistent benefits.** To isolate the benefits of MeCo, we compare it against boundary-centric methods under their respective settings. As shown in Tab. 4, MeCo consistently outperforms the original methods under the same setting across all tasks. Details of the experiments in Tab. 4 are provided in the Appendix D.

**The necessity of both holistic and localized understanding.** As shown in Tab. 5, optimizing the structural token grounding loss without transition tokens (`<tst>`), with segments derived via thresholding, yields significantly poorer performance than when `<tst>` is used. Notably, `<ent>` tokens begin to take effect once query-focused captioning (QFC) is introduced. However, replacing QFC with an uninformative query-copying task (Guo et al., 2025b) reduces performance to the level achieved using only `<ent>` tokens. By combining holistic structural information via `<tst>` tokens with localized details from QFC, MeCo achieves the best performance.

**Sufficient negative samples matter in $\mathcal{L}_{ST}$.** We now discuss the observation that adding a "symmetric" loss term with $p(\mathbf{s}_i|\boldsymbol{h}_t)$ for $\mathcal{L}_{ST}$ (Eq. (4)) led to performance drops, as mentioned in Sec. 3.4. In $p(\boldsymbol{h}_t|\mathbf{s}_i)$ (Eq. (5)), $\boldsymbol{h}_t$ is a *frame*-level feature and $\mathbf{s}_i$ is intended to capture a *segment*. Taking the softmax over frames involves significantly more terms in the summation of Eq. (5) than taking it over the structural tokens, *e.g.*, 100 frames might correspond to only 3 structural tokens. Thus, the losses over $p(\boldsymbol{h}_t|\mathbf{s}_i)$ and $p(\mathbf{s}_i|\boldsymbol{h}_t)$ are not "symmetric" in terms of the softmax. Fewer terms in the denominator of Eq. (5) imply fewer negative samples in Eq. (4), which could negatively affect contrastive learning (He et al., 2020), as shown in Tab. 6 (row 2). To highlight the influence of negative samples, we replace the frame-level features in $p(\boldsymbol{h}_t|\mathbf{s}_i)$ with segment-level features by using $p(\boldsymbol{h}_i^{seg}|\mathbf{s}_i)$, where $\boldsymbol{h}_i^{seg} = \frac{1}{t_i^e - t_i^s}\sum_{t=t_i^s}^{t_i^e}\boldsymbol{h}_t$. This leads to significantly fewer negative samples and drastically reduces performance (row 3-4).

**Boundary-centric methods fail to leverage query-focused captioning.** As shown in Tab. 7, various methods that focus solely on boundary timestamp generation fail to exploit the rich semantic cues provided by QFC. While the former focuses on phase-in and phase-out changes, the latter em-

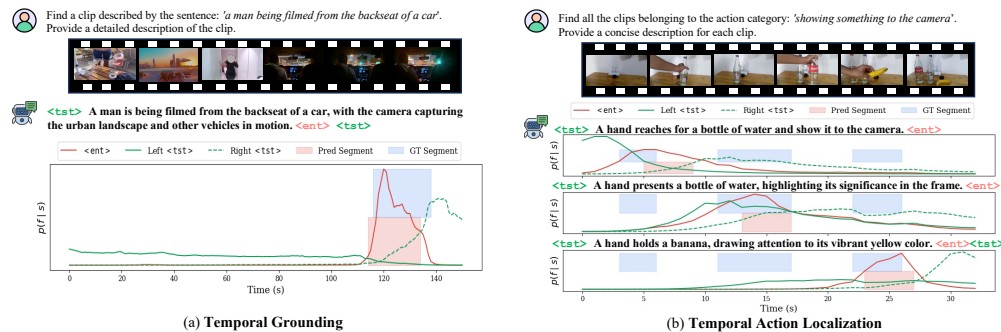

Figure 4: Qualitative examples from temporal grounding and temporal action localization tasks, which involve both single-segment and multi-segment scenarios.

phasizes the most relevant semantic cues. In contrast, MeCo's structural tokens effectively leverage this detailed semantic information to enhance performance.

**Qualitative analysis.** As shown in Fig. 4, MeCo can generate detailed query-focused captions and accurately localize event segments in both single-event and multi-event scenarios. However, there remains considerable room for improvement as there still exist predicted windows that are prominently off from the ground-truth. We provide the visualization of failure cases in Appendix Sec. F.

## 5 CONCLUSION

In this work, we provide a novel perspective of utilizing a semantic-based approach, in contrast to previous boundary-centric methods, to enable video LLMs to handle temporal localization tasks. Instead of directly fine-tuning the video LLMs to generate boundary timestamps, we propose a semantic-oriented framework, MeCo, to better leverage LLM's pre-trained semantic retrieval capability for temporal localization tasks. MeCo is equipped with structural token generation to capture holistic video structures and query-focused captioning tp extract fine-grained event semantics. Facilitated by the structural token grounding module, MeCo can perform a holistic temporal segmentation of the video, readily yielding the timestamps of the queried event segments. Extensive experiments have proven the effectiveness of MeCo on a suite of temporal localization tasks in a zero-shot setting. MeCo also achieves impressive performance in dataset-wise fine-tuning setting for temporal grounding and highlight detection tasks.

## 6 LIMITATION AND FUTURE WORK

Despite the impressive effectiveness of MeCo, we observed that it has a relatively lower boost in fine-grained grounding metrics, *e.g.*, $R@1_{0.7}$. Intrinsically, MeCo prioritizes capturing semantic differences between query-relevant and background frames rather than modeling fine-grained phase-in and phase-out boundary patterns. This reflects an inherent trade-off between semantic-oriented strategies that enable strong zero-shot generalization and boundary-focused modeling that excels at fine-grained localization. Thus, we do not claim to totally replace boundary-centric approaches, which inherently enjoy decent compatibility with the generative modeling of LLMs and can directly model the patterns of segment boundaries. Exploring ways to integrate the strengths of both worlds is a promising avenue for future work. Moreover, we believe that the proposed components are by no means the only options for a semantic-oriented approach.

ACKNOWLEDGEMENTS

This work was supported by JST ASPIRE Grant No. JPMJAP2502 and JST FOREST Grant No. JP-MJFR2160.

REPRODUCIBILITY STATEMENT

To enhance the reproducibility of our work, we made the following efforts. Firstly, we provided detailed steps of the training data synthesis process in Sec. 3.2 with visual aid in Fig. 3 and in the appendix Sec. C and Sec. G. Secondly, we provided detailed pseudo code to facilitate the understanding of the inference process in Sec. E. Thirdly, we provided concrete implementation details in the appendix Sec. B and the annotated code base in the supplementary material, which also contains the generated supervised fine-tuning data in the "data/" folder. Finally, we provided all the evaluation instructions in the appendix Sec. G as well the evaluation code in the supplementary material.

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

# A   THE USE OF LARGE LANGUAGE MODELS (LLMS)

During the drafting and revision of this manuscript, we used GPT5 and Claude Opus 4.1 for checking grammatical errors, formatting the LaTex code for tables and figures, and getting suggestions on academic writing.

# B   IMPLEMENTATION DETAILS

Following E.T.Chat, we perform three stages of training. Throughout all three stages, the visual encoder remains frozen while the frame compressor and projection layer are trainable. In stage 1, we additionally freeze the Q-Former and LLM. In stage 2, we unfreeze the Q-Former and train the LLM with LoRA. In stage 3, we conduct fine-tuning by freezing part of the Q-Former, initializing a new LoRA module for the LLM, and initializing two MLP projectors for the frame and structural token hidden states, respectively, for the structural token grounding module. Following (Liu et al., 2024c), the structural token hidden states are extracted from the second-to-last LLM layer and the frame hidden states from the last layer. Table 8 provides the detailed hyperparameter setup used for stage-3 training. All training was performed on 4 NVIDIA A100-80G GPUs.

We develop MeCo using the E.T.Chat architecture (Liu et al., 2024c), which employs a pre-trained ViT-G/14 from EVA-CLIP (Fang et al., 2023b) as the visual encoder and a resampler consisting of a pre-trained instruction-conditioned Q-Former (Li et al., 2023) followed by a frame compressor (Liu et al., 2024c) that produces one token per video frame. A projection layer projects visual tokens into LLM inputs. The QWen2 base LLM (Wang et al., 2024b) is from MiniCPM-V-2.6 (Yao et al., 2024). We also implemented MeCo with QWen2VL-7B Wang et al. (2024b) by mounting two newly initialized projects from ETChat. For pre-training, we format the structural token sequence based on the temporal order of event and transition segments. For dataset-wise fine-tuning, we observed that the temporal order of structural tokens may inherit the dataset-specific biases in the temporal positions of ground-truth windows (Otani et al., 2020). Therefore, we relax the format of the structural token squence by simply appending a single `<tst>` token to the end which attends to all the transition frames during training and inference.

Table 8: Hyperparameters for stage-3 training.

| MLP Projectors | |
| --- | --- |
| Number of Layers | 2 |
| Hidden Size | 1536 |
| Output Size | 3072 |
| **Large Range LoRA** | |
| LoRA $r$ | 128 |
| LoRA $\alpha$ | 256 |
| LoRA Dropout | 0.05 |
| LoRA Modules | QVO Layers |
| **Model Training** | |
| Max Number of Tokens | 2048 |
| Number of Epochs | 1 |
| Batch Size | 2 |
| Learning Rate for LoRA | 5e-5 |
| LR Decay Type | Cosine |
| Warmup Ratio | 0.03 |
| Optimizer | AdamW |
| AdamW $\beta_1, \beta_2$ | 0.9, 0.997 |

## C  QUERY-FOCUSED CAPTIONING

Based on the temporal localization data in E.T.Instruct (Liu et al., 2024c), we extract event segments and input them to a video captioning model, MiniCPM-V-2.6 (Yao et al., 2024), to generate detailed captions. As these captions often contain redundant information, we summarize them using GPT-4o-mini (OpenAI, 2024) under the condition that the final captions are more detailed than the original localization queries. The annotation pipeline and representative QFC examples are shown in Fig. 5

## D  ADAPTING TIMECHAT AND VTGLLM TO WORK WITH MECO

In Tab. 4, we show the results of integrating MeCo into the TimeChat (Ren et al., 2024) and VT-GLLM (Guo et al., 2025b) models. TimeChat and VTGLLM share the same architecture, with a ViT-G/14 from EVA-CLIP (Fang et al., 2023b) as the visual encoder, a pre-trained Q-Former (Li et al., 2023) as the visual resampler, and VideoLLaMa (Zhang et al., 2023) as the base video LLM. The key difference is that TimeChat applies a sliding video Q-Former to compress the visual tokens to 96, while VTGLLM applies a slot-based visual compressor to obtain 256 tokens. Both use 96 as the maximum number of sampled frames.

To integrate MeCo into this architecture, we modify the video Q-Former in TimeChat to a standard image Q-Former, which resamples 32 tokens to 1 token per frame. Additionally, we apply bidirectional self-attention to the visual token components, following (Liu et al., 2024c). Other components remain unchanged. We then train TimeChat, VTGLLM, and MeCo (adapted) on E.T.Instruct using the same hyperparameters as in (Ren et al., 2024).

## E  MECO INFERENCE

We provide the pseudo code for MeCo's inference process in Algorithm 1. Specifically, after the LLM finishes its generation, we first extract all the structural tokens from the LLM's generated token indices and extract the indices of the event tokens (`<ent>`) `ent_idces` in the extracted structural token list. We then obtain the frame embeddings `fr_emb` and the structural token embeddings `st_emb` by extracting the LLM hidden states and feeding them into the MLP projectors. The pair-wise cosine similarities between the two sets of embeddings are then calculated, which can then be normalized by a `softmax()` operation along the temporal axis to obtain the conditional probability `p_hs` in Eq. (5). Each frame is then assigned to the structural token that leads to be highest conditional probability using the `argmax()` operation. For each `<ent>` token, we find all the frames that have been assigned to it, merge temporally consecutive frames into segments by `split_at_gaps`, and add the obtained segments into `ent_segs` (we observed that sometimes the LLM tends to represent multiple semantically-relevant segments by a single `<ent>`). The start and end timestamps for each segment are simply the timesatmps of the first and last frames in the segment. We do not conduct post-proessing for cases where the segment timestamps do not follow the appearance order of their corresponding `<ent>` tokens. The above pseudo code assumes the most common case where all the event segments are non-overlapping for a clean demonstration of the inference process. In the actual implementation, we process each `<ent>` token one by one and assign frames to each `<ent>` token after suppressing its adacent `<ent>` tokens' (if any) frame scores to take care of overlapping cases.

## F  FAILURE CASES

In Fig. 6, we visualize several typical falure cases: (a) the `<ent>` token attends to the totally wrong segment; (b) the `<ent>` token attends to the correct semantic information but the boundaries are not well estimated by the `<tst>` tokens; (c) the model only generates one events while there are actually three; (d) the video starts directly with the event segment while the model considers the first few frames as transition frames and thus generates a `<tst>` token at first.

**Algorithm 1** Pseudocode of MeCo Inference.

```
# fr_emb: frame embeddings (TxC)
# st_emb: structural token embeddings ((M+K)xC)
# tau: temperature
# ent_idces: indices of event tokens <ent>

# Calculate conditional probabilities (Eq.(5))
zfr = l2_normalize(fr_emb)
zst = l2_normalize(st_emb)
sim = matmul(zfr, zst.T) # Tx(M+K)
p_hs = softmax(sim / tau, dim=0) # Eq.(5)

# Assign frames to structural tokens
assign = p_hs.argmax(dim=1)

ent_segs = []
for i, idx in enumerate(ent_idces):
    # Get indices where assign == idx
    indices = where(assign == idx)

    # Split at discontinuities to get segments
    segs = split_at_gaps(indices)

    # Get start and end timestamps
    timestamps = [[seg[0], seg[1]] for seg in segs]

    # Add to the segment list
    ent_segs.extend(segs) # [[ts_1, te_1], [ts_2, te_2], ...]
```

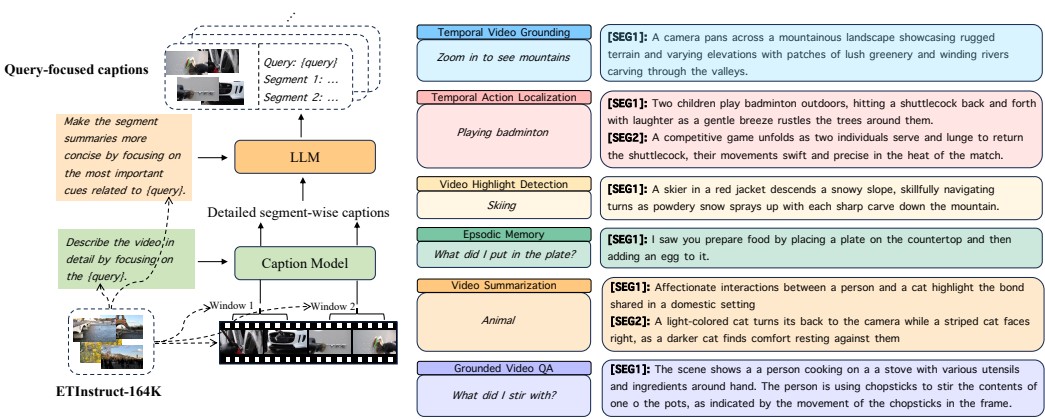

Figure 5: Query-focused captioning pipeline and examples.

## G    EVALUATION AND TRAINING PROMPT TEMPLATES

For evaluation, we modify E.T.Bench templates to work with MeCo. Example templates are provided in Figure 7. For training, we manually craft a query-focused captioning-aware instruction template for each task domain in E.T.Instruct and diversify it with GPT-4o (OpenAI, 2024) to generate four additional templates. The instruction templates for all domains are provided in Fig. 8.

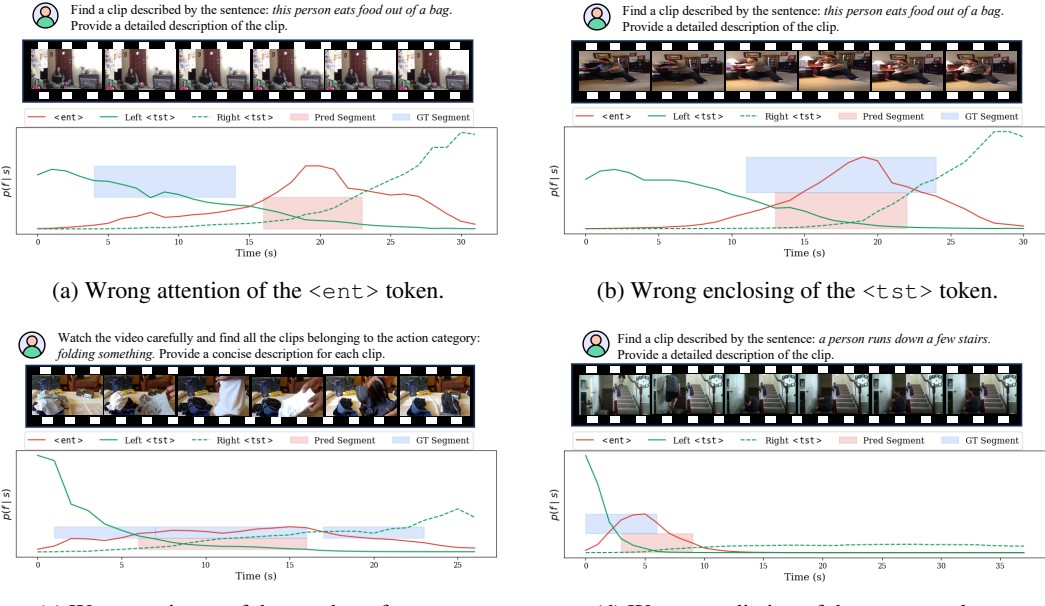

(a) Wrong attention of the `<ent>` token.

(b) Wrong enclosing of the `<tst>` token.

(c) Wrong estimate of the number of events.

(d) Wrong prediction of the `<tst>` token.

Figure 6: Visualization of figure cases.

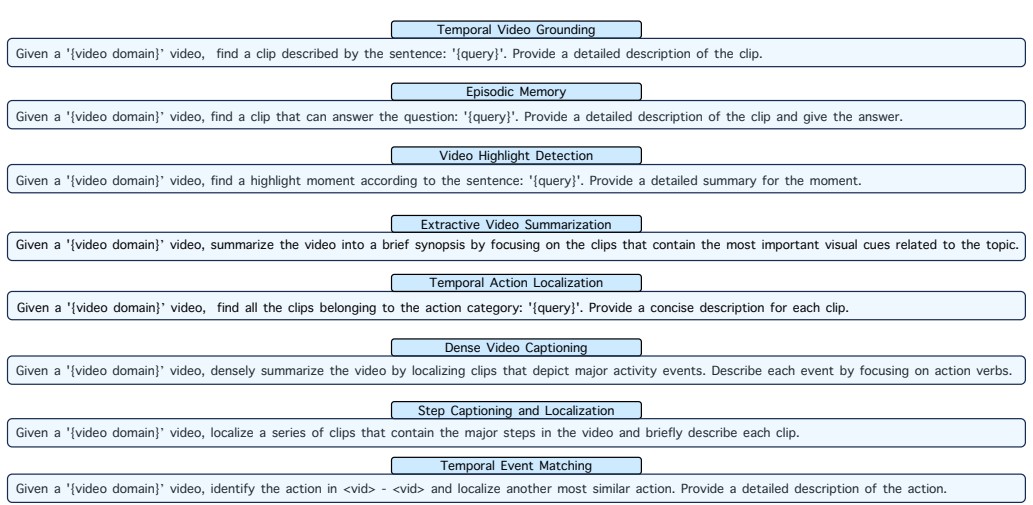

Figure 7: Evaluation prompt templates.

**Temporal Video Grounding**

- Localize the clip described by the given textual query '{query}' in the video and provide a more detailed description of the clip.
- Identify the clip that semantically matches the given textual query '{query}' in the video and provide a description with more relevant temporal and spatial details.
- Given a video and a textual query '{query}', find the clip described by the query in the video and describe it with more important details."
- Localize the clip related to the textual query '{query}' in the video and enhance the description by adding more temporal and spatial details relevant to the query.
- Find the video clip corresponding to the textual query '{query}' and enrich the query with additional but relevant temporal and spatial details.

**Video Highlight Detection**

- Find a representative clip in the video that contains the highlight moment depicted by the given query '{query}' and provide a description to help the user understand what is happening in the highlight moment
- Identify a representative highlight clip in the video that corresponds to the query '{query}' and provide a detailed description rich in important visual activities for the user to better understand the moment.
- Given a video and a query '{query}', find a representative highlight clip in the video that corresponds to the query and describe it with necessary details related to the query for better understanding.
- Find representative highlight clip in the video that corresponds to the query '{query}' and enrich the query with more relevant temporal and spatial details to facilitate better understanding.
- Localize a representative highlight clip in the video that corresponds to the query '{query}' and describe it by focusing on highlight-worthy details to enhance the user's experience.

**Extractive Video Summarization**

- Summarize the video into a brief synopsis by focusing on the clips that contain the most important visual cues related to the video domain '{query}'.
- Identify the clips in the video that contain the most important visual cues related to the video domain '{query}' and provide a concise synopsis of the video.
- Given a video and a video domain '{query}', summarize the clips that contain the most important visual cues to generate a concise synopsis of the video.
- Summarize the video into a concise synopsis by focusing on the clips with the most important visual cues related to the video domain '{query}'.
- Summarize the video into a concise synopsis by localizing the most important clips with visual cues related to the video domain'{query}'.

**Temporal Action Localization**

- Localize all the clips related to the human action '{query}' and provide a concise description for each of them.
- Identify all the clips that depict the human action '{query}' in the video and provide for each of them a concise description.
- For the given action '{query}', localize all the corresponding clips in the video and describe each of them by focusing on action-related visual activities.
- Extract all the clips in the video that depict the action '{query}' and provide a concise description of each clip.
- Given a video and an action '{query}', find all the clips in the video that depict the action and concisely describe each clip.

(a) Temporal grounding.

**Dense Video Captioning**

- Densely summarize the video by localizing clips that depict a series of major activity events. For each clip, first provide a detailed description and then give a brief summary.
- Densely summarize the video's storyline by describing the clips containing major events in temporal order. For each clip, provide a detailed description followed by a corresponding brief summary.
- Capture and concisely describe the events in the video in a dense manner. Provide a detailed description for each event and then summarize it briefly.
- Identify the clips that contain the major events in the video and briefly describe each in temporal order. Provide a detailed description for each clip followed by a brief summary.
- Provide dense captions of the video by providing a sequence of (detailed clip description, brief clip summary) pairs for the major events.

**Step Localization and Captioning**

- Localize a series of clips that contain the major steps in the video. First provide a detailed explanation for each step and then provide a step summary.
- Summarize the video by localizing clips that depict a series of major steps. For each step, provide a detailed explanation followed by a brief summary.
- Identify and localize a series of steps occurring in the video. For each step, provide a detailed explanation and then summarize it briefly.
- Locate the clips involving a sequential series of steps in the video. For each step, provide a detailed explanation followed by a brief summary.
- Summarize the major steps in the video. Provide a detailed explanation for each step followed by a brief summary.

(b) Dense video captioning.

**Grounded Video Question Answering**

- Watch the video carefully with the question '{query}' in mind and find the clip that contains the answer. Provide your analysis of the clip that supports your answer and finally provide the answer itself.
- Identify the clip in the video that helps answer the question '{query}' and provide the answer.
- Given a video and a question '{query}', find the clip in the video that helps answer the question, tell me how you find the answer in the clip, and finally provide the answer.
- Localize the clip in the video that contains the context needed to answer the question '{query}', navigate through the clip and provide the answer to the question.
- Watch the video, find the clip that contains the answer to the question '{query}', analyze the clip and provide the answer.

**Grounded Video Question Answering**

- Densely capture all the clips that construct the storyline of the video. For each clip, provide a detailed description and then summarize it briefly.
- Densely list all the events in the video. For each event, provide a detailed description and then summarize it briefly.
- Densely identify all the events in the video. For each event, provide a detailed description and then summarize it briefly.
- Densely summarize the video by focusing on the major events. For each event, provide a detailed description and then summarize it briefly.
- Densely identify the clips that depict the major events in the video. For each clip, provide a detailed description and then summarize it briefly."

(c) Complex reasoning.

Figure 8: Instruction templates for different task domains: (a) temporal grounding, (b) dense video captioning, and (c) complex reasoning.

