# OpenReview forum: "Measure Twice, Cut Once: A Semantic-Oriented Approach to Video Temporal Localization with Video LLMs"
_ICLR.cc/2026/Conference — ICLR 2026 Poster_

### Official Review · Reviewer_NFTY · 2025-11-01

**Soundness:** 3
**Presentation:** 2
**Contribution:** 2
**Rating:** 4
**Confidence:** 4

**Summary:**

The paper proposes MeCo, a semantic-oriented approach to temporal localization with Video LLMs. Instead of directly predicting boundary timestamps, MeCo first produces structural tokens, which contain an event token (<ent>) and a transition token (<tst>), to segment a video. Before each <ent>, the model inserts an event caption to enrich semantics. During training, each structural token is grounded to its corresponding temporal segment via a contrastive objective. Experiments across grounding and dense captioning tasks show consistent gains over timestamp-centric baselines.

**Strengths:**

1. **Effective grounding objective.** The contrastive grounding loss (Eq. 4) is well-aligned with temporal localization and empirically appears to improve boundary precision.
2. **Strong zero-shot generalization.** Trained on E.T.-Instruct data, MeCo reports competitive or superior zero-shot performance across grounding and dense captioning, indicating good transfer without task-specific fine-tuning.
3. **Ablations support design choices.** The ablation study isolates the contributions of structural tokens and event captions, showing both components are necessary for the full effect.

**Weaknesses:**

1. **Missing implementation details.** The paper presents an inference approach but does not fully specify how timestamps are derived from selected frames or how invalid sequences are handled (e.g., if the index chosen for the first <ent> exceeds that of the second <ent>, whether re-ordering or smoothing is applied).
2. **Limited comparisons to recent methods.** The paper primarily compares with earlier methods (e.g., TimeChat, E.T.Chat). ** More recent temporal grounding approaches, such as DisTime, LLaVA-MR, UniVTG, or Mr.BLIP, are not included. On Charades-STA in particular, several of these report strong results. The paper should either compare these approaches or discuss why MeCo lags behind them.
3. **Title–method mismatch.** The metaphorical title “Measure Twice, Cut Once” does not clearly convey the core technical contribution (semantic structural tokens + contrastive grounding), which could be confusing for readers.

**Questions:**

Please see the weakness part. Particularly, what were the reasons for omitting recent baselines (e.g., DisTime, LLaVA-MR, UniVTG, Mr.BLIP)?

---

> ### Author Response · Authors · 2025-11-24
> **Author Response: Part [1/3]**
>
> We thank the reviewer for the time and effort dedicated to evaluating our work. We appreciate the constructive and insightful feedback. We address the reviewer's concerns below.
> >**[W1] Missing implementation details**. The paper presents an inference approach but does not fully specify how timestamps are derived from selected frames or how invalid sequences are handled (e.g., if the index chosen for the first \<ent\> exceeds that of the second \<ent\>, whether re-ordering or smoothing is applied).
>
> **Timestamp Derivation**: After selecting frames for each event, we merge the temporally consecutive frames into segments. For example, if frames with timestamps [1s, 2s, 3s, 4s, 10s, 11s] are selected for an event, we merge them into two segments: [1s, 2s, 3s, 4s] and [10s, 11s]. Each segment’s start/end times are the first and last timestamps in that segment, e.g, [1s, 4s] and [10s, 11s].
>
> **Invalid Segment Handling**: During evaluation, the invalid segments appear only when there are multiple segments for a single event while the downstream tasks (e.g., grounding, epsodic memory) require only a single segment for each query. In this case, we keep the segment with the highest frame-level score and discard the rest as invalid.
>
> The cases where the frames selected for the first \<ent\> token exceed those for the second \<ent\> token do happen, which can hurt performance for tasks requiring correct temporal order (e.g., dense video captioning). We intentionally did not apply any post-processing such as re-ordering or smoothing so the evaluation reflects the model’s true performance.
>
> We have added these clarifications on timestamp derivation, invalid segment handling, and out-of-order cases to the Appendix E.

---

> ### Author Response · Authors · 2025-11-24
> **Author Response:  Part [2.A/3]**
>
> >**[W2] Limited comparisons to recent methods.** The paper primarily compares with earlier methods (e.g., TimeChat, E.T.Chat). ** More recent temporal grounding approaches, such as DisTime, LLaVA-MR, UniVTG, or Mr.BLIP, are not included. On Charades-STA in particular, several of these report strong results. The paper should either compare these approaches or discuss why MeCo lags behind them.
>
> >**[Q1]** Particularly, what were the reasons for omitting recent baselines (e.g., DisTime, LLaVA-MR, UniVTG, Mr.BLIP)?
>
> ### DisTime (ICCV2025, Oct 2025.)
> We became aware of DisTime only after the ICCV 2025 accepted papers list was posted after Oct 19, which is after the ICLR submission deadline (Sep 24). We include the comparisons below to address the concern.
>
> The pre-training dataset for DisTime contains significantly more event labels for temporal grounding compared to ours, (1656k (DisTime) vs 126k (E.T.Instruct)) and the authors have not released the full pre-training dataset yet. For fair comparisons, we use their released code to train DisTime on E.T.Instruct like us, and evaluate the model's zero-shot performance on ETBench, Charades-STA and QVHighlights.
>
> **ETBench Results**
>
> | Model                     | TVG (F1) | EPM (F1) | TAL (F1) | EVS (F1) | VHD (F1) | SLC (F1) | SLC (Sim) | TEM (Rec) | GVQ (Rec) |
> |--------------------------|---------:|---------:|---------:|---------:|---------:|---------:|----------:|----------:|----------:|
> | DisTime (InternVL2.5 8B) | 58.1     | 3.8      | 16.6     | 30.0     | 60.9     | 27.1     | 14.4      | 23.1      | 3.6       |
> | MeCo (ETChat 3.8B)       | 59.1     | 11.2     | 32.6     | 33.2     | 66.9     | 27.3     | 15.7      | **23.6**  | 9.6       |
> | MeCo (ETChat 7B)         | **62.5** | 15.4     | **35.1** | 35.1     | 66.3     | **30.1** | **16.5**  | 19.1      | 9.9       |
> | MeCo (QWen2VL 7B)           | 59.0     | **17.5** | 34.2     | **35.5** | **67.9** | 28.1     | **16.5**  | 15.4      | **15.1**  |
>
> Note: DVC has been removed as InternVL2.5 has been pretrained on the training set of ETBench DVC samples.
>
> We can see that MeCo models with different base video LLMs deliver consistently better performance across a suite of localization tasks compared to DisTime.
>
> **Chardes-STA and QVHighlights Results**
> | Charades-STA       | $\text{R@1}_{0.3}$ | $\text{R@1}_{0.5}$ | $\text{R@1}_{0.7}$ |
> |--------------------|-------------------:|-------------------:|-------------------:|
> | DisTime            | **71.1**           | 46.5               | 19.9               |
> | MeCo (ETChat 3.8B) | 66.7	              | 44.4	            | 17.5             |
> | MeCo (ETChat 7B)   | 69.6            | 	46.4	              | 19.1               |
> | MeCo (QWen2VL 7B)  | **71.1**           | **50.1**           | **23.3**           |
>
> | QVHighlights (ETBench subset)  | $\text{R@1}_{0.3}$ | $\text{R@1}_{0.5}$ | $\text{R@1}_{0.7}$ |
> |--------------------|-------------------:|-------------------:|-------------------:|
> | DisTime            | 77.3               | 47.8               | 18.4              |
> | MeCo (ETChat 3.8B) | 79.2               | 65.6              |  36.3               |
> | MeCo (ETChat 7B)   | 81.5               | **68.8**               | **41.4**          |
> | MeCo (QWen2VL 7B)  | **82.4**               | 58.2              | 29.8              |
>
> Comparisons on the above grounding datasets prove the superiority of MeCo as well. Note that we provide QVHighlights results based on ETBench TVG samples because DisTime trained on E.T.Instruct failed to handle the multi-segment samples in the original QVHighlight validation set.

---

> ### Author Response · Authors · 2025-11-24
> **Author Response: Part [2.B/3]**
>
> **Continued from Part [2.A/3]**
>
> ### UniVTG (ICCV2023), Mr.Blip (now Chrono-BLIP, arXiv 2025), LLaVA-MR (arXiv 2024)
> We have cited UniVTG and Mr.Blip in our manuscript, though we were not aware of the work LLaVA-MR.  We did not make comparisons with UniVTG and Mr.BLIP because they focus on dataset-wise fine-tuning setting while we focus on pre-training -> zero-shot generalization. Upon revisiting their papers, we found that UniVTG also provided zero-shot results and that Mr.BLIP has upgraded to Chrono-BLIP by including several zero-shot results. Therefore, we provide comparisons in both settings in the following.
>
> **Zero-shot Results**
> | Charades-STA                    | $\text{R@1}_{0.3}$ | $\text{R@1}_{0.5}$ | $\text{R@1}_{0.7}$ |
> |---------------------------|-------------------:|-------------------:|-------------------:|
> | UniVTG                    | 44.1              | 25.2              | 10.0              |
> | Chrono-GPT (GPT4o)        | -                  | 28.8              | 11.0             |
> | Chrono-BLIP (BLIP2 4B)    | -                  | 34.9              | 18.0              |
> | MeCo (ETChat 3.8B)       |  66.7	              | 44.4	            | 17.5              |
> | MeCo (ETChat 7B)          |69.6               | 46.4               | 19.1               |
> | MeCo (QWen2VL 7B)         | **71.1**           | **50.1**           | **23.3**           |
>
> The above results show that MeCo models deliver better zero-shot performance.
>
> Moreover, we have actually implemented ETChat with the interleaved timestamp strategy from Chrono-BLIP, pre-trained it on E.T.Instruct, and made comparisons with MeCo as an ablation study in Table 7 (Row 2, interleaving), the results are as follows:
> | ETBench                         | $\text{F1}_\text{gnd}$ | $\text{F1}_\text{cap}$ | $\text{Sim}_\text{cap}$ | $\text{Rec}_\text{com}$ |
> |---------------------------------|-----------------------:|-----------------------:|------------------------:|------------------------:|
> | Interleaving (Chrono-BLIP)      | 27.1                   | 31.8                   | 15.9                    | 12.4                    |
> | Structural Tokens (MeCo w.t. QFC) | **38.1**             | **33.8**               | **20.5**                | **14.5**                |
>
> The above results also show MeCo's superiority over Chrono-BLIP in terms of zero-shot transfer after large scale pre-training.
>
> **Fine-tuning setting**
>
> When trying to reproduce Chrono-BLIP results with their codebase, we found that its performance varies a lot with hyper-parameters, e.g., batch size, tokenizer pre-processing, etc, and we failed to reproduce their reported results potentially due to differences in CUDA driver and some packages (we did not succeed in installing the recommended environment.) Therefore, for a fair comparison, we report the officially reported results, our reproduced results, and results from MeCo (Chrono-BLIP) based on Chrono-BLIP's codebase for the Charades-STA fine-tuning setting:
>
> |Charades-STA                     | $\text{R@1}_{0.3}$ | $\text{R@1}_{0.5}$ | $\text{R@1}_{0.7}$ |
> |----------------------------|-------------------:|-------------------:|-------------------:|
> | UniVTG                     | 72.6               | 60.2               | 38.6               |
> | LLaVA-MR (w.t. IFS\&DTC)   | -                  | 68.5               | 47.3               |
> | Chrono-BLIP                | -                  | 69.3               | **49.3**           |
> | Chrono-BLIP (reproduced)   | 79.1               | 68.3               | 46.9               |
> | MeCo (Chrono-BLIP)         | **83.6**           | **70.1**           | 45.3               |
>
> where we removed the IFT\&DTC components in LLaVA-MR as they are orthogonal to timestamp representation design.
>
> The results show that MeCo has better performance in terms of less strict tIoU threshold, e.g., $\text{R@1}\_{0.3}$ and $\text{R@1}\_{0.5}$, while struggling with more precise retrieval, e.g., $\text{R@1}\_{0.7}$. This is because MeCo is designed to capture the most semantically relevant part of an event via contrastive learning, which may miss out on the fine-grained transitions near event boundaries. Therefore, there exists trade-offs between leveraging semantic-oriented strategies for better zero-shot generalization and exploiting dataset-wise boundary patterns with timestamp generation. Integrating both worlds could be an interesting future direction. We have also included LLaVA-MR into our Related Work section.

---

> > ### Author Response · Authors · 2025-11-24
> > **Author Response: Part [3/3]**
> >
> > >**[W3] Title–method mismatch**. The metaphorical title “Measure Twice, Cut Once” does not clearly convey the core technical contribution (semantic structural tokens + contrastive grounding), which could be confusing for readers.
> >
> > We appreciate the advice on our title choice. The metaphorical phrae "Measure Twice, Cut Once" intends to convery that we leverage LLM to **Measure both** the global structure of the video via structural tokens and the fine-grained event semantics via the query-focused captions, and finally perform the **Cut Once** via the contrastive grounding to obtain all the segments.
> >
> > However, based on the reviewer's constructive suggestion, we will try to avoid causing readers confusion by making the subsequent part "A Semantic-Oriented Approach to Video Temporal Localization with Video LLMs" more indicative of our technical contributions, which can also echo better with the metaphorical phrase "Measure Twice, Cut Once".

---

### Official Review · Reviewer_Sykj · 2025-11-01

**Soundness:** 2
**Presentation:** 2
**Contribution:** 2
**Rating:** 4
**Confidence:** 3

**Summary:**

The paper introduces MeCo, a framework for temporal localization that replaces timestamp generation with structural tokens: event <ent> and transition <tst>. It augments these tokens with query-focused captioning (QFC), generating detailed captions before each <ent> to refine event semantics. A contrastive structural token grounding module aligns tokens with corresponding video frames to achieve holistic temporal segmentation. Trained on E.T.Instruct along with QFC data, MeCo has strong zero-shot performance across grounding, dense captioning, and complex reasoning tasks.

**Strengths:**

I think the paper has proposed a new shift from timestamp prediction to semantic segmentation with the help of structural tokens (<ent>, <tst>) plus query-focused captioning (QFC).

Also, the author present some results under self conducted experiments with fair comparisons to demonstrate the effectiveness of the proposed method.

**Weaknesses:**

However, I have some severe concern about the paper:

1. Unfair comparison between the proposed method and previous methods. Since the author utilized extra model (MiniCPM-V-2.6) to build the training set, dense captioning will bring more labels to the training set. Thus compared to those timestamp prediction models, it is naturally that the proposed method will have better results. Thus, I think the author should compare previous models with their best results, rather than some complex settings.

2. Critical comparison are missing including TimeChat and TimeChat-T series model. Also, for Charades-STA and QVHighlights, What if the proposed method also fine-tuned on target dataset, rather than the E.T. Instruct dataset? I think the performance gap will be minor or none.

3. the method shows smaller gains for action-focused queries, and the paper acknowledges this limitation without a concrete explanation.

**Questions:**

Please mainly see the weaknesses section above. In general I will give a borderline reject rating.

---

> ### Author Response · Authors · 2025-11-24
> **Author Response: Part [1/3]**
>
> We thank the reviewer for the time and effort dedicated to evaluating our work. We appreciate the recognition of our method as a meaningful contribution to the domain and the constructive feedback provided. We address the reviewer's concerns below.
>
> >**[W1]** Unfair comparison between the proposed method and previous methods. Since the author utilized extra model (MiniCPM-V-2.6) to build the training set, dense captioning will bring more labels to the training set. Thus compared to those timestamp prediction models, it is naturally that the proposed method will have better results. Thus, I think the author should compare previous models with their best results, rather than some complex settings.
>
> We believe there is a misunderstanding regarding both our captioning process and the comparisons. We clarify them as follows.
>
> ### We did not introduce any additional ground-truth event labels
> The reasons are as follows:
> 1. We **did not perform dense captioning** on the training set. We only caption the **original ground-truth event windows** in E.T.Instruct, which **does not introduce any new event (\<ent\>) windows/timestamp labels**. Figure 5 in the Appendix visualizes the captioning pipeline. After the captioning, the **number of ground-truth event windows remains identical** to that of the original dataset.
>
> 2. The transition segments (\<tst\>) are obtained by simply **taking the complementary regions of the ground-truth event windows**. In another word, the event segments are **annotated foreground** while the transition segments are simply the complementary **background segments**.
>
> 3. **No captions or additional queries are generated** for the transition segments (Figure 3 and Eq. (2)). During training, the model is always fed with an **event query provided by the original E.T.Instruct dataset** and learns to distinguish foreground events from background transitions. Therefore, the **localization signal** always comes from the **original event queries and their ground-truth windows**, and transition segments merely serve to convert the localization problem to a binary classification problem, i.e., \<ent\> or \<tst\>.
>
> Threfore all the E.T.Instruct-pretrained models, including ours, use **exactly the same set of queries and ground-truth event windows.**.
>
> ### MeCo achieves the best performance compared to each model's best results.
>
> Though we only boldfaced results pre-trained on E.T.Instruct for comparisons under a controlled setting, we still achieve the best overall performance. The comparisons are as follows:
>
> | ETBench           | TVG (F1) | EPM (F1) | TAL (F1) | EVS (F1) | VHD (F1) | DVC (F1) | DVC (Sim) | SLC (F1) | SLC (Sim) | TEM (Rec) | GVQ (Rec) |
> |------------------|---------:|---------:|---------:|---------:|---------:|---------:|----------:|---------:|----------:|----------:|----------:|
> | Previous best    | 44.3     | 11.1     | 31.8     | 29.7     | 66.9 | **44.3** | **25.3**  | 28.3     | **18.1**  | 19.2      | 3.7       |
> | MeCo best | **62.5** | **17.5** | **35.5** | **35.1** | **67.9**     | 43.4     | 22.8      | **30.1** | 16.5      | **23.6**     | **15.1**   |
>
> Though we consider such a comparison **unfair to us**, as some methods leveraged significantly more training data and compute, MeCO still achieves superiority across the overall tasks.
>
> Despite the misunderstanding, we are gratified to see that the reviewer also emphasizes fair comparisons, which echoes with our efforts to reproduce many methods' results under a controlled E.T.Instruct-based setting.

---

> > ### Author Response · Authors · 2025-11-24
> > **Author Response: Part [2.A/3]**
> >
> > >**[W2]** Critical comparison are missing including TimeChat and TimeChat-T series model. Also, for Charades-STA and QVHighlights, What if the proposed method also fine-tuned on target dataset, rather than the E.T. Instruct dataset? I think the performance gap will be minor or none.
> >
> > ### Comparison with TimeChat and TimeChat-T (VideoChat-T?)
> > **Comparisons with TimeChat.** In the original manuscirpt, we have already compared with TimeChat in both Table 1 (ETBench) and Table 2 (Charades-STA and QVhighlights.) using both its official pre-trained checkpoint and the checkpoint fined-tuned on ET-Instruct using its official codebase. We quote the results below:
> >
> >
> > | ETBench | TVG (F1) | EPM (F1) | TAL (F1) | EVS (F1) | VHD (F1) | DVC (F1) | DVC (Sim) | SLC (F1) | SLC (Sim) | TEM (Rec) | GVQ (Rec) |
> > |-------------------------------------|---------:|---------:|---------:|---------:|---------:|---------:|----------:|---------:|----------:|----------:|----------:|
> > | TimeChat (7B, Official Checkpoint)      | 26.2     | 3.9      | 10.1     | 29.1     | 40.5     | 16.6     | 12.5      | 5.6      | 9.2       | 18.0      | 1.5       |
> > | TimeChat (7B, E.T.Instruct Pre-trained 7B) | 24.3     | 2.3      | 17.7     | 23.6     | 43.0     | 39.4     | 16.5      | 18.0     | 11.8      | 19.1      | 0.8       |
> > | MeCo (ETChat 3B)                    | 59.1     | 11.2     | 32.6     | 33.2     | **66.9** | **43.4** | 20.3      | 27.3     | 15.7      | **23.6**  | 9.6       |
> > | MeCo (ETChat 7B)                    | **62.5** | **15.4** | **35.1** | **35.1** | 66.3     | 43.4     | **20.7**  | **30.1** | **16.5**  | 19.1      | **9.9**   |
> >
> >
> > | Charades-STA                  | $\text{R@1}_{0.5}$ | $\text{R@1}_{0.7}$ |
> > |--------------------------------|-------------------:|-------------------:|
> > | TimeChat (7B, Official results)    | 32.2               | 13.4               |
> > | TimeChat (7B, E.T.Instruct Pre-trained) | 24.9         | 9.2                |
> > | MeCo (ETChat 3.8B)             | 44.4               | 17.5               |
> > | MeCo (ETChat 7B)               | **46.4**           | **19.1**           |
> >
> >
> > | QVHighlights | $\text{mAP}$ | $\text{HIT@1}$ |
> > |-------------------------------------|-----------------------:|---------------:|
> > | TimeChat (7B, Official results)     | 14.5                   | 23.9           |
> > | TimeChat (7B, E.T.Instruct Pre-trained) | 16.4               | 30.6           |
> > | MeCo (ETChat 3.8B)                  | 39.2                   | 61.8           |
> > | MeCo (ETChat 7B)                    | **39.5**               | **64.3**       |
> >
> > **Comparisons with TimeChat-T (VideoChat-T?).** We could not find the method named TimeChat-T, though we realized that we did neglect a model called VideoChat-T [1]. In the following, we will provide the comparisons with VideoChat-T, but would be grateful if the reviewer could provide the reference of TimeChat-T if it is indeed what the reviewer has meant.
> >
> > | Charades-STA                 | $\text{R@1}_{0.3}$ | $\text{R@1}_{0.5}$ | $\text{R@1}_{0.7}$ |
> > |-----------------------|-------------------:|-------------------:|-------------------:|
> > | VideoChat-T (7B)           | 69.9               | 48.7               | **24.0**           |
> > | MeCo (ETChat 7B)      | 69.6               | 46.4               | 19.1               |
> > | MeCo (QWen2VL 7B)     | **71.1**           | **50.1**           | 23.3              |
> >
> >
> > | QVHighlights            | $\text{mAP}$ | $\text{HIT@1}$ |
> > |-----------------------|-----------------------:|---------------:|
> > | VideoChat-T           | 26.5                   | 54.1           |
> > | MeCo (QWen2VL 7B)     | 37.2                  | 57.9          |
> > | MeCo (ETChat 7B)      | **39.5**               | **64.3**       |
> >
> > The above results show that MeCo deliver consistently better results for both benchmarks. Especially on QVHighlights, MeCo has a significant margin compared to VideoChat-T. We provide analysis for why MeCo slightly lags behind in terms of $\text{R@1}_{0.7}$ on Charades-STA in our response **Part [3/3]**.
> >
> >
> > *[1] Timesuite: Improving mllms for long video understanding via grounded tuning. ICLR 2025.*

---

> ### Author Response · Authors · 2025-11-24
> **Author Response: Part [2.B/3]**
>
> **Continued from Part [2.A/3]**
>
> ### Dataset-wise fine-tuning results
> We now provide the results for models fine-tuned on the training sets on Charade-STA and QVHighlight, including both specialist temporal grounding models and video LLM-based models.
>
> | Model              | Charades-STA | Charades-STA        | Charades-STA        | QVHighlights | QVHighlights |
> |--------------------|:------------:|:-------------------:|:-------------------:|:------------:|:------------:|
> |                    | $\text{R@1}_{0.3}$ | $\text{R@1}_{0.5}$ | $\text{R@1}_{0.7}$ | $\text{mAP}$ | $\text{HIT@1}$ |
> | **Specialist models** |            |                     |                     |              |              |
> | M-DETR             | 65.8         | 52.1                | 30.6                | 35.7         | 55.6         |
> | UMT                | -            | -                   | -                   | 39.9         | 64.2         |
> | QD-DETR            | -            | 57.3                | 32.6                | 39.1         | 63.0         |
> | CG-DETR            | 70.4         | 58.4                | 36.3                | 40.8         | 66.7         |
> | UniVTG             | 72.6         | 60.2                | 38.6                | 38.8         | 61.8         |
> | **Video LLMs**     |              |                     |                     |              |              |
> | HawkEye (7B)           | 72.5         | 58.3                | 28.8                | -            | -            |
> | TimeChat (7B)          | -            | 46.7                | 23.7                | 21.7         | 37.9         |
> | VTG-LLM (7B)         | -            | 57.2                | 33.4                | -            | -            |
> | TRACE   (7B)          | -            | 61.7                | 41.4                | -            | -            |
> | VideoChat-T (7B)    | 79.4         | 67.1                | **43.0**            | 27.0         | 55.3         |
> | MeCo (ETChat 3.8B) | 75.3         | 61.6                | 38.5                | 44.6         | 71.8         |
> | MeCo (ETChat 7B)   | 77.2         | 63.9                | 40.1                | 44.7         | 74.3         |
> | MeCo (QWen2VL 7B)  | **82.3**     | **68.5**            | 41.6                | **45.3**     | **75.1**     |
>
> Compared to the zero-shot performance, MeCo models have achieved significant improvements after dataset-wise fine-tuning. Meanwhile, the performance gap between previous works and MeCo variants remain large on the QVhighlight dataset. On Charades, MeCo retains better performance compared to VideoChat-T in terms of $\text{R@1}\_{0.3}$ and $\text{R@1}\_{0.5}$. Though MeCo falls short a bit in terms of $\text{R@1}\_{0.7}$ compared to VideoChat-T, which we explain in our response **Part[3/3]**, it remains competitive and better than all the other models.
>
> The above results deliver another important message: MeCo is the only method that achieves a decent balance between atomic action queries (Charades-STA) and queries with complex semantics (QVHighlights). We thank the reviewer for pointing out the dataset-wise fune-tuning setting, which provides us the opportunity to prove the superior effectiveness of MeCo on both the zero-shot setting and the fine-tuning setting. We have included the above fine-tuning results into Table 2, and provided explanations of them in Section 4.3.

---

> > ### Author Response · Authors · 2025-11-24
> > **Author Response: Part [3/3]**
> >
> > >**[W3]** The method shows smaller gains for action-focused queries, and the paper acknowledges this limitation without a concrete explanation.
> >
> > We appreciate the reviewer's reminder on this point as we get to have an opportunity to ellaborate on it, which we failed to do due to the page limit before submission.
> >
> > Intrinsically, MeCo prioritizes capturing semantic differences between query-relevant and background frames rather than modeling fine-grained phase-in and phase-out boundary patterns. For action-focused queries such as “a person drinks water,” the stages of the process—picking up the cup, lifting it, drinking, and putting it down—often share similar semantics (e.g., "a person holding a cup"). Thus it might be challenging to distinguish such fine-grained transitions based on semantic similarities. As a result, MeCo achieves strong performance at $\text{R@1}\_{0.3}$ and $\text{R@1}\_{0.5}$ by accurately attending to relevant semantics, but shows relatively smaller gains at $\text{R@1}\_{0.7}$. This reflects an inherent trade-off between semantic-oriented strategies that enable strong zero-shot generalization and boundary-focused modeling that excels at fine-grained localization. Exploring ways to integrate the strengths of both directions is a promising avenue for future work. We have incorporated this explanation into the Limitations section.

---

### Official Review · Reviewer_HVDc · 2025-11-01

**Soundness:** 2
**Presentation:** 3
**Contribution:** 3
**Rating:** 6
**Confidence:** 4

**Summary:**

The paper introduces MeCo (Measure Twice, Cut Once), a semantic-oriented framework for video temporal localization using Large Language Models (LLMs). It proposes a conceptual shift from previous methods that rely on generating uninformative boundary timestamps, which often fail to fully leverage the LLM's pre-trained semantic understanding capabilities.MeCo fine-tunes a video LLM using a combination of two generative and one discriminative learning task:Structural Token Generation: A generative task where the LLM partitions the input video based on the user query by outputting a sequence of new, special structural tokens.Query-Focused Captioning (QFC): A second generative task that requires the LLM to produce a detailed caption for each queried event segment immediately. Structural Token Grounding: A discriminative task, implemented using a contrastive learning objective, that maps the semantic information encoded in the structural tokens to the corresponding video segments. Extensive experiments demonstrate that MeCo consistently outperforms timestamp-centric approaches across diverse temporal localization tasks.

**Strengths:**

1.The paper is written with good clarity.
2.The motivation is clear
3. The framework is well-presented

**Weaknesses:**

1.The structural token generation relies on Supervised Fine-Tuning (SFT) labels that require a contiguous, non-overlapping segmentation of the entire video, achieved by augmenting Ground Truth (GT) event boundaries with neighboring transition segments ($\langle tst \rangle$ tokens) to cover the full duration. This presupposes the existence of exhaustive, high-quality, segment-level annotations for the whole video.
2.  The current asymmetric Structural Token Grounding loss introduces a significant imbalance: the number of negative frame samples for a structural token $s_i$ is $T$ (total frames), whereas the number of negative structural token samples for a frame $h_t$ is only $M+K$ (total segments)28. This massive disparity in negative sample cardinality could lead to the loss function being overly dominated by the frame-level discrimination task.

**Questions:**

na

---

> ### Author Response · Authors · 2025-11-24
> **Author Response: Part [1/2]**
>
> We thank the reviewer for the time and effort dedicated to evaluating our work, and we appreciate the recognition of both the quality of the manuscript and the contributions of our proposed method. We address the reviewer's concerns as follows.
>
> >**[W1]** The structural token generation relies on Supervised Fine-Tuning (SFT) labels that require a contiguous, non-overlapping segmentation of the entire video, achieved by augmenting Ground Truth (GT) event boundaries with neighboring transition segments ( tokens) to cover the full duration. This presupposes the existence of exhaustive, high-quality, segment-level annotations for the whole video.
>
> We believe there may be a misunderstanding regarding the annotation of the transition segments.
>
> First, the transition segments are generated solely by inverting the ground-truth event annotations **without requiring any additional annotations for them.** For example, given a 10-second video with a ground-truth event segment [5, 8], the timestamps for the transition segments will simply be [0, 4], and [9, 10]. During training, the model learns to classify segments as event or transition based **only on the annotations of the event segments**, since this forms a straightforward binary setting. We also do not generate Query-focused Captions for transition segments. Therefore, the exhaustive, high-quality, and segment-level annotations are not presupposed to conduct the SFT.
>
> The annotations we use, i.e., E.T.Instruct, actually cover diverse supervision forms: single-segment and multi-segment tasks, annotations of varying quality (both human-curated and automatically produced), and different annotation formats (segment-level and point-level, which we convert to segment-level). As a result, our framework is inherently flexible with respect to the types and granularity of annotations.
>
> We provide the above clarification based on our understanding of the reviewer’s comment, and we would greatly appreciate any correction if we have misinterpreted it. We are glad to offer additional explanations if needed.

---

> > ### Author Response · Authors · 2025-11-24
> > **Author Response: Part [2/2]**
> >
> > >**[W2]** The current asymmetric Structural Token Grounding loss introduces a significant imbalance: the number of negative frame samples for a structural token  is  (total frames), whereas the number of negative structural token samples for a frame  is only  (total segments)28. This massive disparity in negative sample cardinality could lead to the loss function being overly dominated by the frame-level discrimination task.
> >
> > We appreciate the reviewer’s detailed understanding of the loss behavior. However, we believe there may be a misunderstanding regarding which loss function is ultimately used in our final model, likely due to insufficient emphasis in the manuscript.
> >
> > As shown in Eq. (4), we only adopted $p(\mathbf{h}\_{t}| \mathbf{s}\_{i})$ in the contrastive loss, where the negative samples are the frames. We represented this loss function as $\mathcal{L}\_{\text{ST}}(p(\mathbf{h}\_{t}| \mathbf{s}\_{i}))$ in Table 6. We put related results below for further explanations and for the reviewer's easier inspection.
> >
> > | Loss Variant                                                   | $\text{F1}_{\text{gnd}}$ | $\text{F1}_{\text{cap}}$ | $\text{Sim}_{\text{cap}}$ | $\text{Rec}_{\text{com}}$ |
> > |-------------------------------------------------------------------------------------|-------------------------:|-------------------------:|--------------------------:|--------------------------:|
> > | $\mathcal{L}_{\text{ST}}(p(\mathbf{h}_t \mid \mathbf{s}_i))$                    | **40.6**                 | **35.4**                 | **20.3**                  | **16.6**                  |
> > | $\mathcal{L}_{\text{ST}}(p(\mathbf{s}_i \mid \mathbf{h}_t))$                        | 23.4                     | 24.1                  |  12.5                 | 13.2                  |
> > | $\mathcal{L}\_{\text{ST}}(p(\mathbf{h}\_t \mid \mathbf{s}\_i)) + \mathcal{L}\_{\text{ST}}(p(\mathbf{s}\_i \mid \mathbf{h}\_t))$          | 39.9                    | 33.4                    | 15.9                      | 15.2                      |
> >
> > In the table, the asymmetric loss mentioned by the reviewer refers to $\mathcal{L}\_{\text{ST}}(p(\mathbf{h}\_{t}| \mathbf{s}\_{i}))+\mathcal{L}\_{\text{ST}}(p(\mathbf{s}\_{i}| \mathbf{h}\_{t}))$. However, as $\mathcal{L}\_{\text{ST}}(p(\mathbf{s}\_{i}| \mathbf{h}\_{t}))$ uses structural tokens as negative samples and there are only a few of them, it causes significant performance drops (row 2). Combining the two terms $\mathcal{L}\_{\text{ST}}(p(\mathbf{h}\_{t}| \mathbf{s}\_{i}))+\mathcal{L}\_{\text{ST}}(p(\mathbf{s}\_{i}| \mathbf{h}\_{t}))$ recovers part of the performance, which could be because the loss term with frame-level negative samples dominate as described by the reviewer. However, it is still infererior to when only $\mathcal{L}\_{\text{ST}}(p(\mathbf{h}\_{t}| \mathbf{s}\_{i}))$ is used.
> >
> > Therefore, we eventually only applied the single loss term $\mathcal{L}\_{\text{ST}}(p(\mathbf{h}\_{t}| \mathbf{s}\_{i}))$ during our training rather than the asymmetric combination. We have further clarified this choice in the revised manuscript (Section 3.4, L278) and thank the reviewer for drawing attention to this point.
> >
> > We would be glad to provide further details if needed.

---

### Official Review · Reviewer_c33n · 2025-11-01

**Soundness:** 3
**Presentation:** 3
**Contribution:** 3
**Rating:** 6
**Confidence:** 5

**Summary:**

This paper presents MeCo, a semantic-oriented framework that adapts video LLMs for temporal localization by replacing direct timestamp prediction with structured generation. MeCo first emits structural tokens—<ent> (events) and <tst> (transitions)—to outline video structure, augments each event with query-focused captions to enrich semantics, and then grounds tokens to segments using a contrastive objective. Evaluated on E.T. Bench and standard temporal localization benchmarks, MeCo delivers strong and consistent performance.

**Strengths:**

1) **Motivation & insight.**
   The paper makes a compelling case that asking LLMs to emit *uninformative numeric timestamps* underutilizes their semantic reasoning. This observation is clearly articulated and convincingly motivates a *semantic-oriented* alternative.

2) **Strong, broad improvements.**
   MeCo delivers substantial gains across diverse tasks—notably **59.1% vs. 38.6% F1 on TVG (E.T. Bench)**—with **consistent improvements across nine tasks**. It is especially effective for **multi-segment** scenarios (e.g., **26.2 mAP vs. 1.5** for E.T.Chat on QVHighlights).

3) **Comprehensive evaluation & fair comparisons.**
   The experiments span multiple domains (grounding, dense captioning, complex reasoning) and benchmark suites, with thorough comparisons against numerous baselines under matched, transparent settings.

**Weaknesses:**

1) **Incremental novelty via integration.**
   The triad—**structural tokens**, **query-focused captions (QFC)**, and **contrastive grounding**—is well motivated but each component echoes established ideas (tokenized structure, CoT-style captioning, CLIP-like contrast). The contribution lies primarily in a **reformulation and clean integration for localization**, rather than a single, fundamentally new algorithmic primitive.

2) **Temporal granularity concerns.**
   Real videos exhibit **richer, nested structures** (actions ↔ sub-actions, scene changes, shot boundaries). A binary **event/transition** scheme may be too coarse: how are **gradual transitions**, **overlapping events**, or **multi-threaded activities** represented?

3) **Caption overhead vs. payoff.**
   Generating and then attending to QFC introduces **non-trivial compute/latency**. How much of the gain is attributable to **QFC** versus **structural tokens** alone? While Table 5 indicates both matter, the **interaction and marginal utility** (A → A+QFC, A → A+QFC+contrast) aren’t dissected in depth.

4) **Failure-mode analysis is thin.**
   Beyond noting “prominently off” cases (Fig. 4), there’s no **systematic breakdown** of when/why the method fails (e.g., heavy occlusion, rapid camera motion, fine-grained action boundaries, dialogue-driven cues). A short taxonomy with **representative examples** would be valuable.

5) **Limited sensitivity studies.**
   Only **β** and **λ** receive analysis (Fig. 4). Important **design knobs**—**NMS threshold** (0.7), **temperature τ**, **projector architecture/width**, **negative sampling strategy**, **frame sampling rate**—are not explored. A small **hyper-sweep and robustness table** would strengthen claims of generality.

**Questions:**

Please refer to the Weaknesses.

---

> ### Author Response · Authors · 2025-11-24
> **Author Response: Part [1/5]**
>
> We thank the reviewer for the time and effort devoted to evaluating our paper and for recognizing the contributions of our work. We appreciate the reviewer’s insightful comments and address the concerns as follows.
>
> >**[W1] Incremental novelty via integration**. The triad—structural tokens, query-focused captions (QFC), and contrastive grounding—is well motivated but each component echoes established ideas (tokenized structure, CoT-style captioning, CLIP-like contrast). The contribution lies primarily in a reformulation and clean integration for localization, rather than a single, fundamentally new algorithmic primitive.
>
> We appreciate that the reviewer finds the main components, i.e., structural tokens, query-focused captions, and contrastive grounding, to be well-motivated and cleanly integrated for localization. However, we would like to further emphasize the novelty of each component and of the overall framework.
>
> **Structural tokens**: While LLMs have been used to describe temporal structure through natural-language video captions, to the best of our knowledge, we are the first to represent a video as a sequence of compact event and transition tokens. This representation captures compressed semantics at the segment level and avoids the redundancy and noise of natural-language captions. We therefore do not view structural tokens as an established idea, nor as an incremental extension of existing work.
>
> **Query-focused captions**: Though CoT-style captions exist in the synthesized QFCs, they were only used for the grounded video QA task. For other tasks, such as grounding, action localization, and video summarization, we did not enforce the format of CoT and chose to only generate straighforward query-focused captions. In past works, the sysnthesis of CoT data requires complex workflow that involves integration of LLMs, various specialist models, hand-crafted heuristics and sophisticated prompting strategies. Moreover, unlike video QA tasks for which there exist such well-established practices for generating CoT data, it is not clear how to efflectively generate such data for temporal localization tasks. We introduce a simple yet effective approach that generates concise query-focused captions without imposing a CoT format, while still retaining the performance benefits that CoT reasoning provides. Therefore, our explorations of how to use natural-language outputs to help temporal localization and our simple but effective query-focused captioning strategy both represent novel contributions rather than incremental adjustments.
>
> **Constrastive grounding**: Although contrastive learning is a well-known technique, prior temporal localization methods typically use it only as an auxiliary loss within specialist architectures equipped with task-specific heads. In contrast, we demonstrate that contrastively trained semantic similarities alone are highly effective for localization when integrated with video LLMs. Our approach uniquely combines the generative capabilities of LLMs with the discriminative strengths of contrastive learning, offering a new perspective on temporal localization.
>
> Finally, we emphasize that these contributions are not isolated. Their synergistic combination forms a fundamentally different paradigm for temporal localization compared to traditional timestamp-prediction frameworks. Therefore, we view the proposed framework as a novel algorithmic direction, rather than an incremental integration.

---

> > ### Author Response · Authors · 2025-11-24
> > **Author Response: Part [2/5]**
> >
> > >**[W2] Temporal granularity concerns.** Real videos exhibit richer, nested structures (actions ↔ sub-actions, scene changes, shot boundaries). A binary event/transition scheme may be too coarse: how are gradual transitions, overlapping events, or multi-threaded activities represented?
> >
> > ### Temporal gradularity
> > The temporal granularity of the event segments are determined by the semantics specified in the input query. Through contrastive learning, we optimize the \<ent\> tokens to precisely attend to only the query-relevant semantics, whether coarse or fine. As a result, our model can smoothly handle a suite of temporal localization tasks covering a diverse range of temporal granularities, ranging from fine-grained action localization to coarse storyline in dense video captioning. Therefore, the binary event/transition scheme, when conditioned on the input query, is sufficiently expressive to handle diverse temporal granularities.
> >
> > ### Gradual transitions, overlapping events, or multi-threaded activities.
> > From our perspective, the common point of gradual transitions and multi-threaded activities is that the events they represent may temporally overlap with each other. Therefore, we provide our explanation revolving this interpretation of the reviewer's concern, i.e., how to represent overlapping events?
> >
> > In fact, the training data preparation, training and inference phases of MeCo have already considered overlapping events, though we described our method for non-overlapping cases by default for clarity in our explanations. We now detail each phase below:
> >
> > 1. **Training data (structural token labels) preparation**. Normally, when two events are separated by a transition segment, the corresponding structural tokens can be formulated as: \<ent\>\<tst\>\<ent\>. We can then formulate the structural tokens to be \<ent\>\<ent\> to represent 1. two **non-overlapping but temporally consecutive events** and 2. two **overlapping events**, as there is no transition segment separating such events. Therefore, the data preparation stage naturally covers the overlapping cases.
> >
> > 2. **Training**. For two events which are represented as \<ent\>\<ent\> in the structural token sequence, we simply order their temporal windows by the start timestamp values. During training, the \<ent\> tokens will be optimized to attend to the frames in their corresponding windows, respectively, which naturally handles overlapping cases.
> >
> > 3. **Inference**: For straighforward demonstration of our inference process, we only showed the pseudo code (Algorithm 1) for the non-overlapping events, which performs the argmax() to uniquely assign frames to all structural tokens at once. In our actual implementation, we process each \<ent\> token one by one and assign frames to each \<ent\> token after suppressing its adacent \<ent\> tokens' frame scores (if any) to take care of overlapping cases.
> >
> > We have added these clarifications regarding overlapping and concurrent events to the revised manuscript (Figure 3, Appendix E.). We thank the reviewer for raising this important point, which helped us further strengthen the presentation of our method.

---

> > > ### Author Response · Authors · 2025-11-24
> > > **Author Response: Part [3/5]**
> > >
> > > >**[W3] Caption overhead vs. payoff**. Generating and then attending to QFC introduces non-trivial compute/latency. How much of the gain is attributable to QFC versus structural tokens alone? While Table 5 indicates both matter, the interaction and marginal utility (A → A+QFC, A → A+QFC+contrast) aren’t dissected in depth.
> > >
> > > We thank the reviewer for pointing this out. We provide more in-depth dissection of the QFC's benefits.
> > >
> > > | ETBench| $\text{F1}_\text{gnd}$ | $\text{F1}_\text{cap}$ | $\text{Sim}_\text{cap}$ | $\text{Rec}_\text{com}$ |
> > > |----------------------------------|-----------------------:|-----------------------:|------------------------:|------------------------:|
> > > | \<ent\>                          | 26.7                   | 15.0                   | 14.2                    | 9.4                     |
> > > | \<ent\> + QFC                | 40.4                   | 32.0                   | 19.9                    | 14.9                    |
> > > | \<ent\> + \<tst\>                | 38.1                   | 33.8                   | 20.5                | 14.5                    |
> > > | \<ent\> + \<tst\> + QFC    | 40.6               | 35.4               | 20.3                    | 16.6                |
> > >
> > > As can be seen above, without QFC, \<ent\> token alone gives very poor performance. After introducing QFC to \<ent\> to trigger the pre-trained semantic retrieval capability of the LLM, the performance is significantly boosted. Introducing \<tst\> can achieve a similar effect. Introducing QFC to the combination of \<ent\> and \<tst\> brings consistent performance boost to all the localization metrics, i.e., $\text{F1}\_\text{gnd}$, $\text{F1}\_\text{cap}$, and $\text{Rec}\_\text{com}$. Note that such metrics have been averaged from multiple sub-tasks from each task domain, which further highlights the consistent benefits of utilizing QFC. In conclusion, though QFC introduces some computational overhead, its consistent benefits for a suite of localization tasks still render it a very attractive tool, just like how CoT can utilize test-time compute to faciliate many diverse tasks. We updated Table 5 to more clearly convery such a dissection in the revised manuscript.

---

> ### Author Response · Authors · 2025-11-24
> **Author Response: Part [4/5]**
>
> >**[W4] Failure-mode analysis is thin**. Beyond noting “prominently off” cases (Fig. 4), there’s no systematic breakdown of when/why the method fails (e.g., heavy occlusion, rapid camera motion, fine-grained action boundaries, dialogue-driven cues). A short taxonomy with representative examples would be valuable.
>
> As heavy occlusions, rapid camera motion, fine-grained action boundaries, and dialogue-driven cues are common and unavoidable challenges for visual-input-only video LLM–based methods that rely on substantial temporal and spatial downsampling for computational tractability, we provide a more detailed analysis of MeCo’s failure modes below:
> - **Incorrect attention of the \<ent\> token.** The \<ent\> may attend to the wrong segment and completely fail to identify frames containing the correct semantic information.
> - **Correct \<ent\> attention but incorrect \<tst\> coverage.** Even when the \<ent\> token correctly attends to the ground-truth frames, we still need \<tst\> to accurately capture the transition frames to finaly come up with the segment. When \<tst\> covers too many semantically relevant frames or does not cover enough transition frames, the performance can drop.
> - **Incorrect estimation of the number of events.**  In multi-event cases, the model may mispredict the number of events, causing errors in event segmentation.
> - **Incorrect prediction of the existence of \<tst\> token.** When the event lies at the very beginning or end of a video, the model is not supposed to output a \<tst\> token in the beginning or the end, otherwise it will cause the loss of semantically relevant frames and thus reduce the performance.
>
> We added the visualizations of the above failure cases in the Appendix F of the revised manuscript.

---

> ### Author Response · Authors · 2025-11-24
> **Author Response: Part [5.A/5]**
>
> >**Limited sensitivity studies.** Only β and λ receive analysis (Fig. 4). Important design knobs—NMS threshold (0.7), temperature τ, projector architecture/width, negative sampling strategy, frame sampling rate—are not explored. A small hyper-sweep and robustness table would strengthen claims of generality.
>
> We appreciate the reviewer's reminder about the hyper-sweep table, which we provide as follows.
>
> **NMS Threshold**
>
> As we only used NMS for the baselines without the \<tst\> token, we report the results obtained from the \<ent\> + QFC baseline.
>
> | Threshold | $\text{F1}_\text{gnd}$ | $\text{F1}_\text{cap}$ | $\text{Sim}_\text{cap}$ | $\text{Rec}_\text{com}$ |
> |-----------|-----------------------:|------------------------:|------------------------:|------------------------:|
> | 0.5       | 39.6                   | 31.9                    | 19.6                    | 14.9                    |
> | 0.6       | 40.1                   | 31.9                    | 19.6                    | 14.9                    |
> | 0.7       | **40.4**               | **32.0**                | **19.9**                | 14.9                    |
> | 0.8       | **40.4**               | 31.7                    | 19.7                    | 14.9                    |
>
> The results show that the NMS has very little effect to the results, but 0.7 is the value that gives the best overall results.
>
> **Temerature $\tau$**
> | $\tau$| $\text{F1}_{\text{gnd}}$ | $\text{F1}_{\text{cap}}$ | $\text{Sim}_{\text{cap}}$ | $\text{Rec}_{\text{com}}$ |
> |-----------------------:|-------------------------:|-------------------------:|---------------------------:|---------------------------:|
> | 0.04                   | 39.5                     | 34.9                     | 19.0                       | 16.1                       |
> | 0.07                   | **40.6**                 | **35.4**                 | **20.3**                   | 16.6                   |
> | 0.10                   | 39.8                     | 35.0                     | 19.4                       | **17.0**                   |
>
> The results show that 0.07 gives the best overall results, though different thresholds in a reasonable range do not cause too much performance variation. We chose 0.07 the our default value.
>
> **Frame sampling rate**
>
> We provide MeCo (ETChat 3.8B) results for this ablation.
>
> | FPS | $\text{F1}_{\text{gnd}}$ | $\text{F1}_{\text{cap}}$ | $\text{Sim}_{\text{cap}}$ | $\text{Rec}_{\text{com}}$ |
> |----:|-------------------------:|-------------------------:|---------------------------:|---------------------------:|
> | 0.5 | 38.5                     | 33.9                     | 20.8                   | 14.8                       |
> | 1   | 40.6                     | **35.4**                 | 20.3                       | 16.6                       |
> | 2   | **41.4**                 | 35.1                     | **20.8**                   | **17.3**                   |
>
> Higher value for sampling fps gives better results. However, consider the computational cost incurred by using FPS=2 and the relatively small performance gains, we used FPS=1 as the default value for the ETChat-based model.

---

> > ### Author Response · Authors · 2025-11-24
> > **Author Response: Part [5.B/5]**
> >
> > **Continued from Part [5.A/5]**
> >
> > **Projector width**
> > | Dim | $\text{F1}_{\text{gnd}}$ | $\text{F1}_{\text{cap}}$ | $\text{Sim}_{\text{cap}}$ | $\text{Rec}_{\text{com}}$ |
> > |----:|-------------------------:|-------------------------:|---------------------------:|---------------------------:|
> > | 768  | 40.5                     | **36.1**                 | **20.8**                   | 14.5                       |
> > | 1536 | **40.6**                 | 35.4                     | 20.3                       | **16.6**                   |
> > | 3072 | 39.8                     | 34.3                     | 19.8                       | 15.7                       |
> >
> > As different widths do not give very different results, we follow ETChat to use 1536 as the default projector width.
> >
> > **Negative sampling strateges**
> >
> > We have provided an ablation of negative sampling strategies in Table 6. For the convenience of the reviewer's inspection, we add more explanations to each row and put the results below.
> >
> > | Negative sampling strategies | $\text{F1}_{\text{gnd}}$ | $\text{F1}_{\text{cap}}$ | $\text{Sim}_{\text{cap}}$ | $\text{Rec}_{\text{com}}$ |
> > |----------------------------------|--------------------------:|--------------------------:|---------------------------:|---------------------------:|
> > | $\mathcal{L}_{\text{ST}}(p(\mathbf{h}_t \mid \mathbf{s}_i))$ (Frames as negative) | **40.6** | **35.4** | **20.3** | **16.6** |
> > | $\mathcal{L}_{\text{ST}}(p(\mathbf{s}_i \mid \mathbf{h}_t))$  (Strcutural tokens as negative)                      | 23.4                     | 24.1                  |  12.5                 | 13.2                  |
> > | $\mathcal{L}\_{\text{ST}}(p(\mathbf{h}\_t \mid \mathbf{s}\_i)) + \mathcal{L}\_{\text{ST}}(p(\mathbf{s}\_i \mid \mathbf{h}\_t))$ (Frames and structural tokens as negative) | 39.9 | 33.4 | 15.9 | 15.2 |
> > | $\mathcal{L}\_{\text{ST}}(p(\mathbf{h}^{\text{seg}}\_i \mid \mathbf{s}\_i))$ (Segments as negative)| 23.2 | 23.5 | 12.9 | 13.4 |
> > | $\mathcal{L}\_{\text{ST}}(p(\mathbf{h}^{\text{seg}}\_i \mid \mathbf{s}\_i)) + \mathcal{L}\_{\text{ST}}(p(\mathbf{s}\_i \mid \mathbf{h}^{\text{seg}}\_i))$  (Segments and structural tokens as negative)| 24.3 | 25.9 | 13.6 | 13.4 |
> >
> > The above results show that using frames alone as the negative samples delivers the best results, as using structural tokens or segment-level features tends to give only a few number of negative samples. Therefore, we chose frame-based negative sampling as the default strategy. We have provided detailed discussion about these results in Section 4.4.

---

### Author Response · Authors · 2025-12-01
**Rebuttal Summary**

Dear ACs, SACs, and PCs,

Thank you for the significant additional effort you are putting into the decision process under the current unusual circumstances. We summarize the reviews and our rebuttal comments below to help reduce the burden on your decision making.

We first summarize the positive feedback as follows:
- **A novel framework**: The proposed method, MeCo, was described as a "new/conceptual shift" from previous methods (@HVDc, @Sykj).
- **Clear motivation**: The proposed method was considered "convincingly/clearly motivated" (@c33n, @HVDc), based on a "clearly articulated" observation (@c33n).
- **Clear presentation**: The paper was noted to be "written with good clarity," and the framework was described as "well-presented" (@HVDc).
- **Strong performance across diverse tasks**: The proposed method delivers strong performance (@c33n: "strong, broad improvements", @HVDc: "consistently outperforms...", @Sykj: "strong zero-shot performance", @NFTY: "consistent gains"), validated through "comprehensive and fair comparisons" (@c33n, @Sykj) "across grounding, dense captioning, and complex reasoning tasks" (@c33n, @HVDc, @Sykj, @NFTY).

The constructive feedback from the reviewers guided us to clarify potential points of misunderstanding, provide additional technical details, and conduct further experiments, which we detail below.

**Clarified Misunderstandings**
- **Novelty (@c33n)**: We clarified that our proposed Structural Tokens, Query-Focused Captioning (QFC), and LLM-based Contrastive Grounding are novel contributions rather than established ideas, and that their integration is also novel.
- **Assumption about exhaustive annotations (@HVDc)**: We clarified that we only require standard temporal localization annotations (with flexible formats), and that transition segments are produced without any additional annotations.
- **Loss function (@HVDc)**: We clarified that the loss in question was only a baseline in the ablation; the main method uses a more robust loss.
- **Captioning pipeline and comparison fairness (@Sykj)**: We clarified that our event-only captioning pipeline uses the original annotations and does not introduce extra event labels or dense-captioning-style supervision, preserving fair comparisons.
- **Title (@NFTY)**: We clarified the intended meaning of “Measure Twice, Cut Once” and will revise the title to more directly reflect the paper’s contributions.

**Additional Technical Details**
- **Temporal granularities and overlapping cases (@c33n)**: We clarified that our framework can flexibly handle diverse temporal granularities and overlapping events, and added corresponding explanations to the revised manuscript.
- **Failure modes (@c33n)**: Following the reviewers’ suggestions, we added failure-mode analyses and visualizations to the revised manuscript.
- **Limitations (@Sykj)**: We provided more detailed explanations of our method’s limitations in fine-grained grounding and incorporated them into the revised manuscript.
- **Implementation details (@NFTY)**: We added clarification on how final timestamps are derived and confirmed that we did not apply the post-processing mentioned by the reviewer.

**Additional Experiments**
- **More detailed ablations of Query-Focused Captioning (QFC) (@c33n)** show clear and consistent gains attributable to QFC.
- **Hyperparameter sweep (@c33n)** shows that the framework is robust across a range of hyperparameter values, as long as they are within a reasonable range.
- **Comparisons with VideoChat-T (@Sykj)** show that our framework outperforms VideoChat-T in both zero-shot and fine-tuning settings on Charades-STA and QVHighlights. We also acknowledge our limitation in fine-grained grounding compared to VideoChat-T.
- **Comparisons under dataset-wise fine-tuning setting (@Sykj)** on Charades-STA and QVHighlights show that the proposed method remains strong in this setting. In particular, we achieve a significant performance boost on QVHighlights compared to prior work.
- **Comparisons to DisTime (ICCV 2025, Oct 19), UniVTG (ICCV 2023), MrBLIP (arXiv 2025), and Llava-MR (arXiv 2024) (@NFTY)**: We first provided comparisons with DisTime on E.T.Bench under a zero-shot setting and showed that our method performs better. We then provided zero-shot comparisons with UniVTG and MrBLIP on Charades-STA and ETBench, which again demonstrate stronger performance of our method. Finally, we compared against UniVTG, MrBLIP (Chrono-BLIP), and Llava-MR on Charades-STA under a fine-tuning setting, and found that we achieve the best R@1 at IoU 0.3 and 0.5, while being slightly behind at IoU 0.7, for which we provided our analysis.

We sincerely thank you again for your invaluable time and effort in evaluating our work.

Best regards,
Authors of Paper 6676

---

### Meta-Review · Area_Chair_dCaX · 2025-12-25

**Summary:**

The major concerns raised by reviewers relate to (i) the novelty of individual components as well as their integration, (ii) assumptions about supervision and fairness of comparisons, (iii) missing comparisons or implementation details in the initial submission, and (iv) limitations in fine-grained action boundary localization. Through an extensive rebuttal, the authors provided clarifications, additional experiments, and new comparisons addressing most technical concerns. While some limitations remain, particularly regarding fine-grained boundary precision and compute overhead, the overall evidence supports the paper as a solid contribution above the acceptance threshold.

**Reviewer Concerns:**

Addressed:
1.  Assumptions about supervision and exhaustive annotations: The authors clarified that transition segments are derived directly from existing event annotations and do not require additional labeling. They further demonstrated that the framework accommodates diverse annotation formats and non-exhaustive supervision.

2. Loss imbalance and grounding formulation: The rebuttal clarified that the asymmetric loss discussed by the reviewer is not used in the final model, and provided ablations showing why the chosen formulation is preferable.

3. Fairness of comparisons and “extra supervision” via captioning: The authors convincingly explained that no additional event labels are introduced via captioning, and that all methods compared under the identical supervision.

4. Missing baselines and recent work: The authors added detailed comparisons with recent methods across both zero-shot and fine-tuning settings.

5. Implementation details: The authors clarified timestamp derivation, handling of invalid or out-of-order segments, and explicitly stated that no post-processing is applied. This addresses reproducibility concerns. Additional ablations and hyperparameter sweeps were provided and demonstrate robustness.

Remained concerns:
1. Incremental novelty vs. integration: While the rebuttal makes a reasonable case that structural tokens and QFC are not straightforward extensions of prior work, the contribution is still best characterized as a novel reformulation and integration rather than a fundamentally new primitive. This is acceptable but limits the perceived depth of novelty.

2. Fine-grained action boundary precision: The authors acknowledge that MeCo underperforms some timestamp-based methods at strict IoU thresholds (e.g., IoU 0.7) on action-centric datasets. The explanation is plausible, but this remains a real limitation rather than a resolved issue.

3. Compute and latency overhead of QFC: The rebuttal demonstrates that QFC provides consistent gains, but the trade-off between performance and computational cost is not fully quantified in practical deployment settings.

**Reviewer Scores:**

Reviewer c33n and HVDc likely keep their positive scores; Reviewer Sykj likely moves from 4 to 5 or 6 as fairness/baseline concerns are addressed; Reviewer NFTY likely moves from 4 to 5 or 6 with added inference details and additional comparisons.

---

### Decision · Program_Chairs · 2026-01-26

Accept (Poster)